# Optimal and learned algorithms for the online list update problem with Zipfian accesses

**Piotr Indyk**                                                                INDYK@MIT.EDU
*MIT*

**Isabelle Quaye**                                        ISABELLEAYOKOR.QUAYE@GMAIL.COM

**Ronitt Rubinfeld**                                              RONITT@CSAIL.MIT.EDU
*MIT*

**Sandeep Silwal**                                                     SILWAL@CS.WISC.EDU
*UW-Madison*

**Editors:** Gautam Kamath and Po-Ling Loh

## Abstract

The online list update problem is defined as follows: we are given a list of items and the cost to access any particular item is its position from the start of the list. A sequence of item accesses come online, and our goal is to dynamically reorder the list so that the aggregate access cost is small. We study the stochastic version of the problem where the items are accessed i.i.d. from an unknown distribution $p$. The study of the stochastic version goes back at least 60 years to McCabe.

In this paper, we first consider the simple online algorithm which swaps an accessed item with the item right before it, unless it is at the very front. This algorithm is known as the Transposition rule. We theoretically analyze the stationary behavior of Transposition and prove that its performance is within $1 + o(1)$ factor of the optimal offline algorithm for access sequences sampled from heavy-tailed distributions, proving a conjecture of Rivest from 1976.

While the stationary behavior of the Transposition rule is theoretically optimal in the aforementioned i.i.d setting, it can catastrophically fail under adversarial access sequences where only the last and second to last items are repeatedly accessed. A desirable outcome would be a policy that performs well under both circumstances. To achieve this, we use reinforcement learning to design an adaptive policy that performs well for both the i.i.d. setting and the above-mentioned adversarial access. Unsurprisingly, the learned policy appears to be an interpolation between Move-to-Front and Transposition with its behavior closer to Move-to-Front for adversarial access sequences and closer to Transposition for sequences sampled from heavy tailed distributions suggesting that the policy is adaptive and capable of responding to patterns in the access sequence.

**Keywords:** reinforcement learning, list update, learned algorithms, online algorithms

## 1. Introduction

The list update problem is one of the most fundamental problems in online algorithms, with its study going back to the very foundations of competitive analysis Sleator and Tarjan (1985). In the list update problem (also called the list access problem or self-organizing sequential search), we maintain a list of $n$ items or records. Requests for records come online, and the cost to service a request is the location of the desired record i.e., the number of records preceding the record in the list, plus one. After serving any request, we can rearrange the list in hopes of minimizing costs for future accesses which are unknown. The goal is to minimize the sum of the costs across all item requests,

compared to an optimal offline algorithm, measured via the competitive ratio (see Section 2 for the formal setup). Thus intuitively, an online algorithm must 'learn' to organize the list against unknown future requests.

In this paper, we are primarily interested in the *stochastic* version where the accesses are generated i.i.d. from an unknown access distribution $p$ over $[n]$. The study of the stochastic version is motivated by the strong lower bounds for the general case Vaze (2023), and the reasonable assumption that requests may not be adversarially generated. In fact, the study of the stochastic version predates the general case and goes back to at least 1965 to McCabe, who analyzed the performance of the classic Move-to-Front (MTF) algorithm under stochastic requests. MTF always moves the requested item all the way to the very front. In the natural case where the access pattern has a heavy-tailed distribution (e.g. we expect some items to be accessed more than others), McCabe's analysis shows that MTF is $\log(4) \approx 1.386$ competitive. This is already better than any algorithm we can hope for the general case, where a competitive ratio lower bound of $1.5$ is known Vaze (2023) .

Later, Rivest in 1976 analyzed both MTF and another algorithm called Transposition in a similar stochastic setting. Transposition swaps the requested record with the record immediately to the left, i.e, it moves the requested record one step forward if possible. Note that in addition to being extremely simple, rules such as MTF and Transposition are computationally inexpensive and require no memory to execute, meaning they 'self-organize' the list. In the stochastic case, Rivest showed that for any access distribution $p$, the expected cost of Transposition is always lower than MTF Rivest (1976). Based on this result, the paper conjectured that Transposition is the optimal algorithm when $p$ follows a Zipfian distribution[1]. The main theoretical contribution of this work is to prove the conjecture. Let $\text{Zipf}(\alpha)$ be the distribution where the $i$th most frequent record is accessed with probability $\propto 1/i^\alpha$. We show:

**Theorem 1.1** *Suppose $p$ follows the Zipf($\alpha$) distribution for $0 < \alpha \leq 2$. In this case, the competitive ratio of Transposition $1 + o(1)$.*

The range $0 < \alpha \leq 2$ captures the most natural regime for the problem; see Section 2.

While theoretically optimal in the aforementioned i.i.d. setting, Transposition can catastrophically fail if the i.i.d. hypothesis is 'severely' violated. Indeed, consider an adversarial access pattern where the last two items in the list are repeatedly requested one after another. Any 'intelligent' algorithm should move these items to the first two positions, yet Transposition fails to do this.

Thus, we are motivated to design an algorithm which can take advantage of any i.i.d. or heavy tailed access patterns as done by Transposition, but also performs well under the adversarial access pattern. There are many methods one can employ. One idea is to switch between MTF and Transposition as we observe a change in the query pattern. Unfortunately, this will require us to maintain frequency counters to learn the distribution over items and detect when that distribution changes. The second option, which is quite natural, is a traditional learning-augmented algorithm that relies on predictions from an oracle. A good candidate for the prediction is where to place a queried item. Presumably, this oracle will be a learned frequency estimator, such as is described in Hsu et al. (2019). A downside of this approach, as with most learning-augmented algorithms, is that performance degrades as the quality of frequency estimates degrades. Instead, if we could extract a policy rather than predictions from an oracle, then in the face of declining performance, we can analyze the

---

1. A simple example on 6 items shows that Transpositions is not the exact optimal for general $p$'s Anderson et al. (1982)

policy's behaviour to understand why. Therefore, we choose a reinforcement learning based approach.

The second contribution of this paper is to demonstrate empirically that a simple reinforcement learning agent that uses deep Q-learning can recognize patterns in request access sequences and adapt its policy to lower the aggregate cost of servicing these requests. From our experiments, we observe the following:

1. The RL agent learns an online list update policy whose performance closely matches the best heuristic-based algorithm when evaluated across different distributions.

2. The RL agent is able to adjust its policy accordingly even when there is a change in the frequency distribution over items in the list or when the list itself changes.

## 2. Preliminaries and Cost Model

The state of a list will be interchangeably thought of as a permutation $\pi$ of $\{1, \ldots, n\} := [n]$, where $\pi(i)$ denotes the position of record $i$. In the stochastic case, $p$ denotes the access distribution, meaning the $i$th item or record is accessed with probability $p_i$, independently across requests. Note that $p$ is unknown to the online algorithm. Without loss of generality, we assume $p_1 \geq \ldots \geq p_n$. For a fixed list $\pi$, the expected cost to service a request is

$$\text{Cost}(\pi) := \sum_{i=1}^{n} p_i \pi(i). \tag{1}$$

In the stochastic case, we use the standard assumption that OPT knows the distribution $p$ in advance, but not the randomness of the draws. Thus, OPT places the items in decreasing order of $p_i$ and never rearranges the list. That is, the $i$th most frequently requested item is in the $i$th position in OPT's list Rivest (1976). By the rearrangement inequality Wikipedia contributors (2024), this minimizes the expected cost of a request drawn from $p$. We denote the cost of OPT as

$$\text{OPT}(p) := \sum_{i=1}^{n} p_i \cdot i. \tag{2}$$

Let $\mathfrak{S}_n$ denote the set of all permutations of $[n]$. In the stochastic setting, MTF and Transposition can be thought of as markov chains where the states are permutations in $\mathfrak{S}_n$ and the respective algorithms determine the transitions between states. For example for MTF, if we are in a particular state $\pi \in \mathfrak{S}_n$, then for every $i$, with probability $p_i$, we transition to a state where record $i$ is moved all the way to the front in $\pi$. For Alg $\in \{\text{MTF, Transposition}\}$, a stationary distribution $Q_{\text{Alg}}$ over $\mathfrak{S}_n$ exists Rivest (1976); Gamarnik and Momčilović (2005). Intuitively, $Q_{\text{Alg}}$ captures the distribution over states that each algorithm is in over the long term.

**Cost Model.** Given the current state of the list, the cost to serve a request for record $r$ is its position $i_r$ in the list. After servicing this request, we are allowed to move $r$ to any index before $i_r$ for free. This models the fact that we have already traversed the list and can remember any pointers to a previous index (e.g. for linked lists). An algorithm can also move record $r$ further back in the list, at the cost of the number of positions beyond $i_r$, but none of the algorithms we consider do this, so this can be disregarded. The standard cost model Vaze (2023) also incorporates moving non-requested

items, but again none of the algorithms we study, as well as OPT (in the stochastic case), perform such actions, so we again ignore this case. For Alg $\in$ {MTF, Transposition}, we define their access cost in terms of the stationary distribution. Intuitively, this can be thought of as their average cost in the long horizon. Their expected cost is defined as

$$\text{Cost}(\text{Alg}, p) := \sum_{\pi \in \mathfrak{S}_n} \text{Cost}(\pi) Q_{\text{Alg}}(\pi). \tag{3}$$

We say that an algorithm Alg has competitive ratio at most $C$ for access distribution $p$ if it holds that $\text{Cost}(\text{Alg}, p) \leq C \cdot \text{OPT}(p)$. McCabe McCabe (1965) showed that for any $p$, we have

$$\text{Cost}(\text{MTF}, p) = 1 + 2 \sum_{1 \leq i < j \leq n} \frac{p_i p_j}{p_i + p_j}.$$

Rivest Rivest (1976) showed that for any $p$, we have $\text{Cost}(\text{Transposition}, p) \leq \text{Cost}(\text{MTF}, p)$.

**Zipfian Access Distribution.** In the Zipf(1) case of $p_i \propto 1/i$, we can use the above formula to compute the competitive ratio of MTF as $\log(4) + o(1) \approx 1.386$[2]. Thus, Rivest also showed that Transposition has competitive ratio at most 1.386. Recall that we let Zipf($\alpha$) be the distribution where the $i$th most frequent record is accessed with probability proportional to $1/i^\alpha$. Lastly, we remark that $0 < \alpha \leq 2$ is the most natural regime for Zipf($\alpha$) for this problem. For $\alpha = 0$, the distribution is uniform meaning *every* permutation has the same expected access cost, so there is nothing interesting to say. For $\alpha > 2$, we have $\text{OPT}(\text{Zipf}(\alpha)) = O(1)$. Now in reality, the pointer operations that are implicit in the problem definition have some fixed additive cost overhead. This cost is overwhelmed and negligible when the OPT $= \omega(1)$, but becomes a non-trivial contribution in the regime $\alpha > 2$. Thus, Zipf($\alpha$) for $\alpha > 2$ is a not a natural access distribution for the problem, so we only consider $0 < \alpha \leq 2$.

Lastly, we note that we only consider 'memory-less' algorithms which take as input the current state of the list and an accessed item, and outputs a new list. These algorithms are also called 'self-organizing' Rivest (1976) and their memory overhead does not depend on the number of accesses. Virtually all of the algorithms studied for the online list update problem fall in this category (both the stochastic and general case) Vaze (2023).

**Other related works** The list update problem has been well studied over the years Burville and Kingman (1973); Hendricks (1972, 1973); Knuth (1997); McCabe (1965); Schay and Dauer (1967); Sleator and Tarjan (1985); Rivest (1976) with many algorithms, both deterministic Sleator and Tarjan (1985); Albers (1998) and randomized Reingold et al. (1994); Albers et al. (1995); Irani (1991); Irani et al. (1991), being put forth. We refer to the book Vaze (2023) for a friendly introduction. Some applications of the list update problem include maintaining a dictionary under space limitations as in Bachrach and El-Yaniv (1997) as well as applications in computational geometry (Bentley et al. (1993))

We remark that works such as Kan and Ross (1980); Tenenbaum and Nemes (1982); Gamarnik and Momčilović (2005) have also studied Transposition under i.i.d. accesses, but their results do not imply our main theorem (Kan and Ross (1980); Tenenbaum and Nemes (1982) do not study Zipfian distributions and the main theorem of Gamarnik and Momčilović (2005) does not imply a bound on the competitive ratio or handle the $\alpha = 1$ case, to the best of our knowledge). The use of machine

---

2. In the paper $o(1)$ refers to a quantity going to 0 as $n \to \infty$.

learning models in algorithms design is a rapidly growing field, often termed learning-augmented algorithms. We refer to https://algorithms-with-predictions.github.io/ for an up-to-date collection of literature on learning-augmented algorithms.

## 3. Theoretical Analysis

Recall our main theorem.

**Theorem 1.1** *Suppose $p$ follows the Zipf($\alpha$) distribution for $0 < \alpha \leq 2$. In this case, the competitive ratio of Transposition $1 + o(1)$.*

The goal of this section is to give a high-level, intuitive, overview of the proof, as well as a proof sketch. All omitted details and proofs are deferred to Appendix A.

### 3.1. Proof Overview and Intuition

The starting point of the proof is to rewrite the expression for the cost of an algorithm. Recall from Equation 3 that $\text{Cost}(\text{Alg}, p) := \sum_{\pi \in \mathfrak{S}_n} \text{Cost}(\pi) Q(\pi)$, where $Q$ denote the stationary distribution of the algorithm over the set of permutations (we omit the dependence on the algorithm). Another equivalent way to write the cost (from Rivest (1976)) is

$$\sum_i p_i \cdot \left( 1 + \sum_{1 \leq j \leq n, j \neq i} b(j, i) \right), \tag{4}$$

where $b(i, j)$ is the probability that record $i$ is before record $j$ in the stationary distribution $Q$. To explain this, note that record $i$ is accessed with probability $p_i$, and the cost to find it is the same as the number of elements to the left of it in the current list. This is captured by the variables $b(i, j)$.

Now the intuition is that if $i < j$ (and thus $p_i \geq p_j$), we want $b(i, j)$, the probability that $i$ is before $j$, to be very large. Ideally, as in OPT, this probability is 1. Note that $b(i, j) + b(j, i) = 1$, so an equivalent way to lower bound $b(i, j)$ is to lower bound the following ratio:

$$\frac{b(i, j)}{b(j, i)} = \frac{\sum_{\pi, \, i \text{ appears before } j} Q(\pi)}{\sum_{\pi, \, j \text{ appears before } i} Q(\pi)}. \tag{5}$$

Again in OPT, this ratio is infinite since $b(j, i) = 0$, and our goal is to show that this ratio is very large for Transposition (if $i < j$). If we are able to lower bound the ratio by a large factor, this automatically translates into a large lower bound for the probability $b(i, j)$, and thus a smaller competitive ratio compared to OPT. To control this ratio, Rivest (1976) show the following key fact for Transposition.

**Lemma 1** *Let $\pi$ be a permutation where element $a$ is immediately before element $b$. Let $\pi'$ be the same permutation but with $a$ and $b$ swapped (applying one swap). Then*

$$\frac{Q_{\text{Transposition}}(\pi)}{Q_{\text{Transposition}}(\pi')} = \frac{p_a}{p_b}.$$

This above lemma is quite intuitive. For example, if $p_a > p_b$, then the permutation that has $a$ before $b$ has a higher probability. This fact is enough to show how Rivest (1976) prove their main result, which is

$$\text{Cost(Transposition}, p) \leq \text{Cost(MTF}, p).$$

It will also be the starting point of our main theoretical result, which shows that Transposition is within $1 + o(1)$ of OPT for Zipfian $p$.

**How to prove Rivest (1976)'s main result.** Note that in fraction 5, there is a $1 : 1$ mapping between the permutations that appear in the numerator and those that appear in the denominator (simply swap $i$ and $j$). Let $\pi$ and $\pi'$ be a pair of such permutations, one appearing in the numerator and its 'match' in the denominator. Suppose $i$ appears $t$ spots before $j$ in $\pi$. Then we have $Q(\pi) = (p_i/p_j)^t Q(\pi')$ by repeatedly applying Lemma 1 (move $i$ all the way to $j$ and then move $j$ all the way back. Everything that is not related to $i$ and $j$ cancels). Since $t \geq 1$, we have $Q(\pi) \geq (p_i/p_j)Q(\pi')$ and thus the fraction 5 implies

$$\frac{b(i,j)}{b(j,i)} \geq \frac{\sum_{\pi,\, j \text{ appears before } i} Q(\pi) \cdot (p_i/p_j)}{\sum_{\pi,\, j \text{ appears before } i} Q(\pi)} \geq \frac{p_i}{p_j}.$$

Equality implies $b(i,j) = p_i/(p_i+p_j)$ (since $b(i,j)+b(j,i) = 1$), which is exactly the corresponding formula for MTF McCabe (1965). Hence Transposition dominates MTF, completing the argument of Rivest (1976).

**Our Proof Intuition.** The prior discussion motivates the following high level idea. If we could *pretend* that the denominator in fraction 5 was only summing over permutations $\pi$ where $j$ and $i$ were speared by $\geq 2$ positions, then we could lower bound the ratio $b(i,j)/b(j,i)$ by $(p_i/p_j)^2$, meaning that we get a correspondingly larger lower bound for $b(i,j)$, directly implying a smaller competitive ratio. More generally, *if* we could ignore permutations in the denominator where $i$ and $j$ are separated by at most $t$ positions, then we have $b(i,j)/b(j,i) \geq (p_i/p_j)^t$. A large $t$ implies $b(i,j) \approx 1$.

Of course in reality, we cant simply ignore such inconvenient permutations. However, this prompts us to show that the 'contribution' of the permutations where $i$ and $j$ are separated by 'few' elements is 'small'. Note that 'contributions' is quantified by the probability mass that $Q$, the stationary distribution, places on these permutations. Thus, we wish to show that the probability of $i$ and $j$ being separated by few places, in the case where $i \ll j$, is very small. To do so, we concoct a novel *charging* argument, which shows that the probability of such undesirable permutations, under $Q$, is dominated by the permutations where $i$ and $j$ are separated by many places.

### 3.2. Setup of the Proof

We assume $n$ is sufficiently large and $0 < \alpha \leq 2$ is a constant. Note that our access distribution is $\text{Zipf}(\alpha)$. Define $D_t$ to be the set of permutations where the following holds. Note that these permutations appear in the denominator of fraction 5.

- $j$ appears before $i$, and

- there are $t$ other elements between them.

For technical reasons, we are interested in the following parameter regime. $i \geq \log n, j \geq i + i^{0.9}$, and $0 \leq t \leq i^{0.15} := T$. Note that we only care about the cases where $j$ is sufficiently larger than

$i$. Otherwise, the ratio $b(i,j)/b(j,i)$ being large is simply not true. For example, even in OPT, switching the positions of say the 5th and the 6th most accessed items does not increase the cost by a noticeable amount. Our parameter choices reflect this. In this range of $t$, let $\pi \in D_t$ and $k = j^{0.75}$. Going back to our intuition, we want to show that $\sum_{0 \le t \le T} Q(D_t)$ is negligible. Towards this end, we partition these permutations as follows:

Case 1 : $= \{$An element $\ge j - 2k$ is immediately to the left of $j\}$,
Case 2 : $= \{$At least half of $j - k, \ldots, j - 1$ are to the left of $j\} \setminus \{$Case 1$\}$,
Case 3 : $=$ The rest.

The above cases are exhaustive. We define an explicit map $f : D_t \to \mathfrak{S}_n$ as follows, depending on which case $\pi \in D_t$ falls in. The map $f$ will map each permutation in $D_t$ to one that is outside of the set, in a way that the probability, under $Q$, does not decreases too much. That is, for each of the permutations in the above cases, we want $Q(f(\pi)) \ge Q(\pi)$. We define $f$ as follows.

- **Case** 1: swap the element immediately to the left of $j$ with $j$. This permutation is $f(\pi)$.

- **Case** 2: Let $e$ denote the element immediately to the left of $j$. We first swap $e$ with the element in $S := \{j - k, \ldots, j - 1\}$ that is leftmost in the current list. Then we swap $j$ with the element in $S$ that is leftmost in the current state of the list (after the first swap). The resulting permutation is defined to be $f(\pi)$.

- **Case** 3: First swap $i$ with $j$. Then swap the element $e$ with $j$ where $e$ is among $j - k, \ldots, j - k/3$ that is in the *right most* position. Again, the resulting permutation is defined to be $f(\pi)$.

As stated before, in all cases, we want to 'charge' the probability of the permutations in these cases to a permutation outside of $D_t$. This is formalized in the lemmas below.

### 3.3. Sketch of Main Analysis

We now argue about the behaviour of $f$ under each case, which also motives the design of the three cases. For all the lemmas in this section, we are working under the parameters as defined in Section 3.2 and $Q$ is the stationary distribution for the Transposition algorithm. All omitted proofs are in Section A. The following three lemmas execute the above charging plan for the three cases.

For the first case, the element directly to the left of $j$ cannot be too small compared to $j$, so swapping it with $j$ should not decrease the probability under $Q$, using Lemma 1. This allows us to insert a another element between $j$ and $i$ after the swap, which we iterate later in Theorem 3.1.

**Lemma 2**
$$\sum_{\pi \in Case\ 1} Q(\pi) \le \frac{p_{j-2k}}{p_j} \cdot \sum_{\pi' \in \{f(\pi) | \pi \in Case\ 1\}} Q(\pi'). \tag{6}$$

In case 2, we also insert many elements between $j$ and $i$. This is because we know know there are many elements in $S = \{j - k, \ldots, j - 1\}$ to the left of $j$ and we swap $j$ with the left most element. This allows us to charge the probability of the current permutation to one where $j$ and $i$ are separated by many places (which is desirable according to our intuition from Section 3.1). However, we are

putting $j$ *before* a record with a higher access probability. This means the resulting permutation can have a much smaller probability under $Q$ (due to Lemma 1). To mitigate this, we *pay* for this swap by first swapping the element (record) directly to the left of $j$. Since we are not in case 1, this element must be smaller than $j - 2k$. Everything in $S$ is significantly larger than this element, meaning that we gain a huge boost in probability if we move this element to the left of *every* element in $S$. This initial swap helps pay for swapping $j$ (and motives our cases).

**Lemma 3**

$$\sum_{\pi \in Case\ 2} Q(\pi) \leq \sum_{\pi' \in \{f(\pi) | \pi \in Case\ 2\}} Q(\pi'). \tag{7}$$

*Furthermore, for any $\pi$ in Case 2, we have $f(\pi) \in D_m$ for $m > T$.*

Lastly for case 3, we know there must be many elements in $S$ to the right of $j$. When we swap $j$ with the right most element in $S$, the resulting permutation has $i$ and $j$ separated by many places, which is very helpful, again going back to our intuition of Section 3.1. Specifically, we show

**Lemma 4**

$$\sum_{\pi \in Case\ 3} Q(\pi) \leq \left(1 - \frac{k}{3j}\right)^{\alpha k/2} \cdot \sum_{\pi' \in \{f(\pi) | \pi \in Case\ 3\}} Q(\pi'). \tag{8}$$

Combining the analysis of the three cases allows us to prove our most technical estimate. It shows that in the regime of Section 3.2, the ratio $b(i, j)/b(j, i)$ is large.

**Theorem 3.1** *Let $b(i, j)$ be the probability that $i$ appears before $j$ under $Q$ and analogously define $b(j, i)$. Under the parameter settings of Section 3.2, we have*

$$\frac{b(i, j)}{b(j, i)} \geq (1 - o(1)) \frac{1}{4T} \cdot \left(\frac{j}{i}\right)^{\alpha T}.$$

**Putting everything together.** Now we briefly sketch how the above analysis proves our main result, Theorem 1.1. Recall that we want to bound cost given by Equation 4, which we split into partial terms as follows:

$$\text{Cost(Alg)} = \sum_{i=1}^{n} p_i \cdot \left(1 + \sum_{1 \leq j \leq n, j \neq i} b(j, i)\right)$$

$$= 1 + \underbrace{\sum_{i=1}^{n} p_i \left(\sum_{1 \leq j \leq i + i^{0.9}} b(j, i)\right)}_{\text{Estimate 1}} + \underbrace{\sum_{i \leq \log n} p_i \left(\sum_{j > i + i^{0.9}} b(j, i)\right)}_{\text{Estimate 2}} + \underbrace{\sum_{i > \log n} p_i \left(\sum_{j > i + i^{0.9}} b(j, i)\right)}_{\text{Estimate 3}}.$$

We provide intuition for bounding each of the estimates and defer the details to Appendix A.

- Estimate 1: It will mostly suffice to use the naive bound that $b(j, i) \leq 1$. This is because we *already expect* any $j < i$ to be before $i$ since $p_j \geq p_i$ (this is how they are arranged in OPT). Overall, Lemma 6 in Appendix A will show the contribution of Estimate 1 is $(1 + o(1))$OPT. This forms the bulk of the cost. (Note OPT $= \omega(1)$).

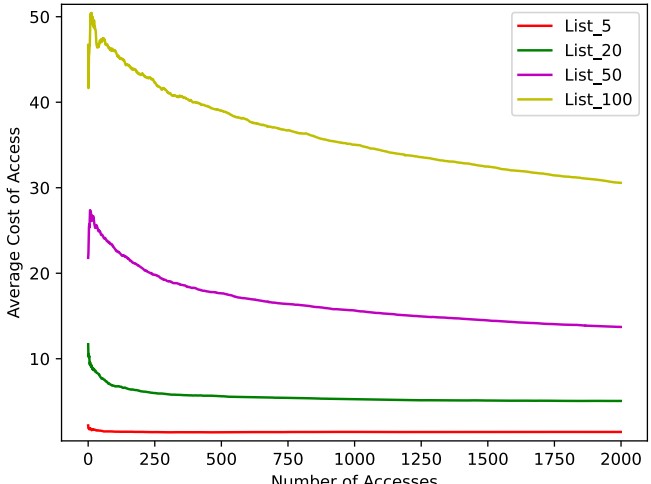

Figure 1: Average cost of first 2000 accesses for Transposition Algorithm on different list sizes.

- Estimate 2: We show that the contribution of Estimate 2 is $o(\text{OPT})$. Here it will suffice to use the simple estimate that $b(i, j) \geq p_i/(p_i + p_j)$ for $i < j$ proved in Rivest (1976) (see the intuition in Section 3.1).

- Estimate 3: This is the most technical estimate which requires the full power of Theorem 3.1. Here, we are under the parameter regime of Section 3.2 and Theorem 3.1 implies a large lower bound for $b(i, j)/b(j, i)$. We show in Lemma 8 that the contribution of this estimate is $o(\text{OPT})$.

Putting together the three estimates proves Theorem 1.1.

Lastly, we remark that while we have focused on the Zipfian case, since it was highlighted in Rivest (1976), we believe our overall charging proof scheme could be applicable to a wider range of access distributions. It is thus tantalizing to ask the following question for future work: Is the competitive ratio of Transposition $1 + o(1)$ for all access distributions $p$? (We know that for a sized $n = 6$ array Transposition is not optimal, but the lower bound of Anderson et al. (1982) does not say anything about large list sizes).

## 4. Reinforcement Learning Agent Design

In the previous section, we showed theoretically that the Transposition algorithm is optimal for access sequences sampled i.i.d. from heavy-tailed distributions such as a Zipf($\alpha$) with $0 < \alpha \leq 2$. Unfortunately, for large lists, the Transposition algorithm can take a long time to converge. Consider Figure 1 where we sample an access sequence of length 2000 from a Zipf(1) distribution and simulate the online list update algorithm on lists of sizes 5, 20, 50 and 100 that are maintained by the Transposition rule. For the same 2000 accesses, the lists of size 50 and 100 do not become stationary.

Additionally, the Transposition algorithm can perform poorly on access sequences that are not sampled i.i.d. For instance, suppose we have an access sequence where only the last and second-

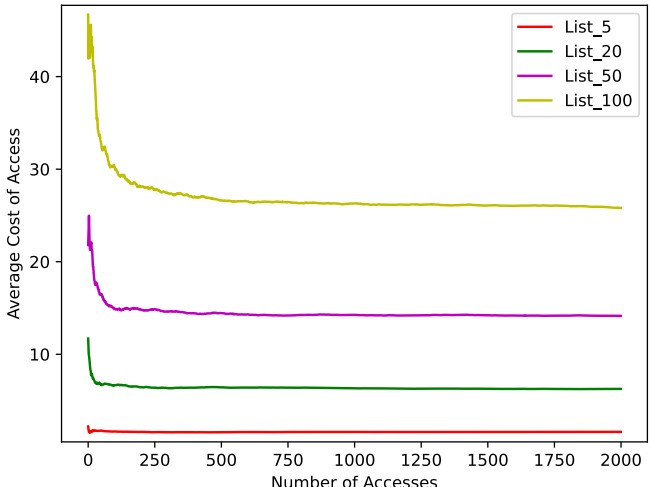

Figure 2: Average cost of first 2000 accesses for Move-to-front Algorithm on different list sizes.

to-last items in the list are accessed.[3]. We call such a sequence an **adversarial access sequence**. For such a sequence, Transposition performs poorly since it keeps those two items in the last two positions of the list when it would be optimal to move them to the front.

The Move-to-Front algorithm, however, performs well under the adversarial sequence and also converges fairly quickly as shown in Figure 2. Thus, although it does not have the same optimal theoretical guarantees as the Transposition rule for Zipf($\alpha$), its quick convergence makes it preferable in practice. In this section, our goal is to see if we can develop a new algorithm or technique which converges relatively quickly and is competitive with the optimal algorithm regardless of whether the access sequence is sampled i.i.d or not. Since we are interested in finding an optimal policy, we choose to use reinforcement learning. We begin our discussion of the reinforcement learning design by describing the list update problem as a Markov Decision Process:

1. Environment: The environment comprises the current list $L$ and the access sequence $R$.

2. State: The state is the list $L$ as in Rivest (1976). The state space is therefore all the possible permutations of the list. Indeed, transitions in this state space are Markovian, since the next permutation is independent of all previous permutations: it is only dictated by the current permutation and the action taken to move an item in the list. An important note here is that this state space has size $O(|L|!)$. Computationally, this presents a challenge. To handle this challenge, we choose to apply a transformation $C$ over the state space to shrink it to a poly-sized one for computational feasibility. The transformation $C$ which we apply, takes as input the current list $L$ and the item requested $r_t$ and returns as the state a tuple of the position of the requested item in $L$, denoted as $L(r_t)$ and the item requested by $r_t$ itself. Our transformation function therefore maps the initial $O(L!)$ state space to an $O(L^2)$ state space. We considered multiple candidates for

---

3. By last and second to last in the list, we mean last and second-to-last in the original ordering of the list before any rearrangements were made

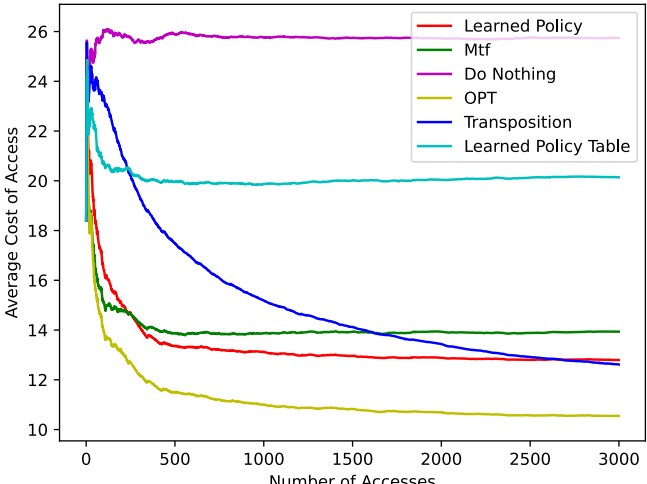

Figure 3: Average cost of first 3000 accesses for different algorithms on a list of size 50.

$C$ and a discussion of them as well as the results that led us to choose the above transformation function can be found in the Appendix B.3.

3. Action: An action is the new position $r_t$ is moved to. Thus, the size of the action space $\mathcal{A}$ is $|L|$. [4] One idea is to restrict $\mathcal{A}$ to positions $l$ such that $0 \le l \le L(r_t)$. However, this limits the family of policies that the agent can learn so we refrain from doing so.

4. Reward Function: At time step $t$ where item $r_t$ is queried, if $r_t$ is found in position $i$ and the agent elects to move it to position $j$, then the agent receives a reward $-2 * \max(j, i)$.

We choose to use deep Q-learning. Recall that we have applied a compression function $C$ over the state space. If we choose to use tabular Q-learning, we will have to approximate the original $L! \times L!$ table with an $L^2 \times L^2$ table which expectedly does not produce as good of a result as deep Q-learning which Figure 3 shows.(Compare the cost curve labelled "Learned Policy" which is the cost curve corresponding to deep Q-learning and the cost curve labelled "Learned Policy Table" which is the cost curve corresponding to tabular Q-learning.) Moreover, as the list size grows, the performance gap between tabular Q-learning and deep Q-learning also grows. We therefore need a powerful function approximator hence our choice of deep Q-learning. Much like in the original deep Q-learning paper, Mnih (2013), we use experience replay and also employ a target network to achieve a more stable learning outcome. For our purposes, an experience is a tuple of a starting state, the action taken in that state, the reward earned for that action and the next state resulting from taking the action.

The architecture for the Q-network can be seen in Table 1[5] and all hyper-parameters used can be found in Table 2.

---

4. Keeping $r_t$ in the same position is also a possible action.
5. We use a one-hot encoding for the input layer

Table 1: Neural Network Architecture

| Layer Type | Number of Units | Activation Function |
| --- | --- | --- |
| Input Layer | $2|L|$ | - |
| Dense Layer | $64 \cdot 8$ | ReLu |
| Dense Layer | $2|L|$ | ReLu |
| Dense Layer | $64 \cdot 8$ | ReLu |
| Dense Layer | $|A|$ | Linear |

Table 2: Choice of reinforcement learning agent parameters

| Name of Hyper-parameter | Description | Value |
| --- | --- | --- |
| Learning rate | Controls the weighting of forecasted future Q-values and current Q-values | 0.7 |
| Epsilon(range of values) | Controls the level of exploration vs. exploitation | 0.01-1 |
| Decay Rate | Controls the rate at which epsilon is update over time | 1.25 |
| Discount factor | How much we discount future rewards | 0.9 |
| Batch size | Controls the size of the window the agent can sample experiences from | 128 |

## 5. Experiments & Results

Before we can get a policy, we must first train the reinforcement learning agent. In our case, we train the reinforcement learning agent on access sequences that fall into one of three categories:

- Zipfian access sequence: This is an access sequence where each request in the sequence is sampled i.i.d. from a Zipf(1) whose support are the keys in $L$.

- Adversarial access sequence: This is an access sequence where only the last and second-to-last items in the table are accessed. Note that last and second-to-last are with respect to the initial ordering of items in the list prior to servicing requests. Throughout this section, when we talk about a non-i.i.d. access sequence, we are referring to this type of access sequence.

- Uniform access sequence: This is an access sequence where each request in the sequence is sampled i.i.d. from a Uniform distribution whose support are the keys in $L$.

In addition to varying the pattern in the access sequence during training, we also vary the list size to verify the agent's behavior as the size of the list is scaled up and down. List sizes are chosen from the set $\{10, 20, 50, 100, 500, 1000\}$. Therefore in total, we have 18 different variations of the experiment we ran. During each of these experiments, we have two phases: the training phase and the testing phase. In the training phase, the reinforcement learning agent is permitted to learn and update its Q-table. In the testing phase, no more learning occurs and we simply exploit the learned policy to understand its behavior and performance. For a given experiment, the access sequences for both the training and testing phase are sampled from the same distribution. Each experiment is also repeated ten times to ensure that the results are consistent. Putting all this together, an experiment follows the following steps:

1. Start by choosing a list size from $[10, 20, 50, 100, 500, 1000]$ and then create a list $L$ of the chosen size whose entries are integers chosen randomly from the range 1 to 100000.

2. Once we have our list of integers $L$, we choose one of three types of access sequences i.e. Zipfian, Uniform or Adversarial. We then sample an access sequence $R$ from the respective distribution with the numbers in $L$ as its support. We split the access sequence into two **unequal** parts. One part which we call $R_{train}$ will be used to train the reinforcement learning agent and the other part which we call $R_{test}$ will be used to test the agent's learned policy for performance. For most of our experiments, $R_{train}$ has a length of 10000 and $R_{test}$ has a length of 3000.

3. We commence the training phase using an epsilon-greedy strategy where each time step follows roughly the following steps:

   - We receive a request $r_t$ from $R_{train}$.
   - We look up $r_t$ in $L$.
   - We decide what action to take i.e. which item to move and where.
   - We perform the action by moving the chosen item to its new place in $L$.
   - We record the reward, get the new state and update Q-table values where necessary.
   - We repeat until we have serviced all requests from $R_{train}$.

4. Once the training phase is over, we enter a testing phase where we exploit the policy learned by the agent thus far. Before we start the testing phase, we shuffle the list $L$ so as to break up any rearrangements from the training phase that would give the agent an advantage. To serve as benchmarks, we also create four lists $L_{MTF}, L_{Trans}, L_{Do-Nothing}$ and $L_{OPT}$ which are identical to $L$. These four lists will also service the requests from $R_{test}$ just like the original list $L$ but the lists will be maintained according to the Move-to-front, Transposition, Do-Nothing[6] and Optimal[7] policies respectively.

5. The testing phase follows roughly the following sequence of steps:

   - We receive a request $r_t$ from $R_{test}$.
   - Each of the five lists $L, L_{MTF}, L_{Trans}, L_{Do-Nothing}, L_{OPT}$ service this request and we record the cost associated with each one.
   - We then perform the necessary rearrangements in each list as dictated by the rspective algorithms. We also record the rearrangements performed by each algorithm.
   - We repeat until we have serviced all requests from $R_{test}$.

6. We repeat this experiment 10 times and construct average cost curves for each algorithm using costs recorded in the testing phase.

## 5.1. Results

We begin by looking at the average cost curve of the learned policy for different access sequences. Unless otherwise stated, we focus on cost curves for lists of size 50. In Figure 4(a), 4(b) and 4(c) we observe that the learned policy's average cost curve is always competitive with either Move-to-Front or Transposition. Even as the list size varies, we see that this competitiveness is maintained as shown

---

6. Do-Nothing means no action is taken when requests are serviced. i.e. the list permutation never changes.
7. Optimal here means an algorithm that maintains frequency counters and orders items by these counters.

in Figure 10. From these cost curves, it is likely the learned policy's behavior changes depending on the access sequence it is servicing. To understand how the policy changes, we examine the rearrangement choices made by the learned policy during the testing phase by constructing a 2D heat map called a **policy map**. We construct the policy map by recording the frequency of each rearrangement choice. So, if an algorithm moves an item that was in position $a$ to position $b$, we increment the frequency counter in the grid cell with coordinates $(a, b)$. We normalize frequency values per column and only use rearrangement choices made in the testing phase to construct the policy maps. None of the results presented here involve numbers recorded in the testing phase. In

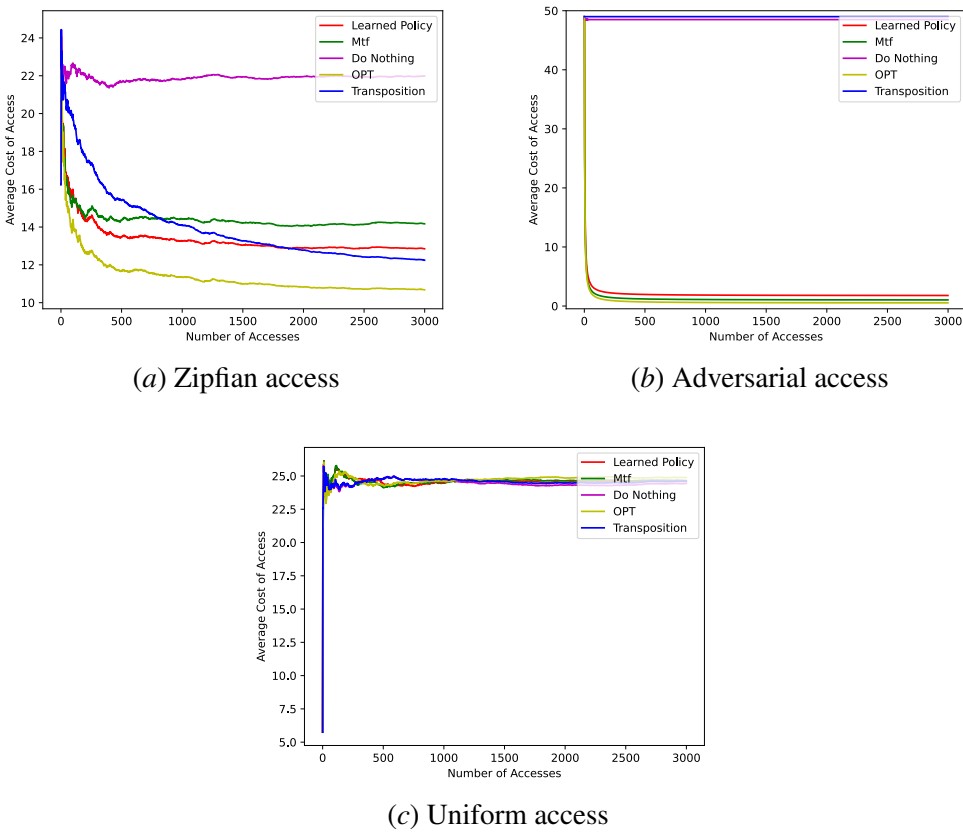

($a$) Zipfian access

($b$) Adversarial access

($c$) Uniform access

Figure 4: Average cost for a list of size 50 on different access sequences.

Figures 5($a$), 5($b$) and 5($c$), we see an example policy map constructed from the rearrangement choices made by the Transposition algorithm on a list of size 10. Since the Transposition algorithm always moves items one position ahead, only the off-diagonal cells have non-zero frequency in the policy map. Likewise in Figures 18($a$), 18($b$) and 18($c$) we see an example policy map constructed from the rearrangement choices made by the Move-to-Front algorithm on a list of size 10. Move-to-Front only moves items to the front of the list and so only the cells at the very top of the grid have non-zero frequency. Unless otherwise indicated, we only show policy maps for lists of size 10.

Turning our attention to policy maps for the learned policy (Figure 6) we observe some off-diagonal behavior as in Transposition, but cells to the upper-right side of each off-diagonal cell also show non-zero frequency. This behavior is replicated across list sizes as seen in Figures 7 and

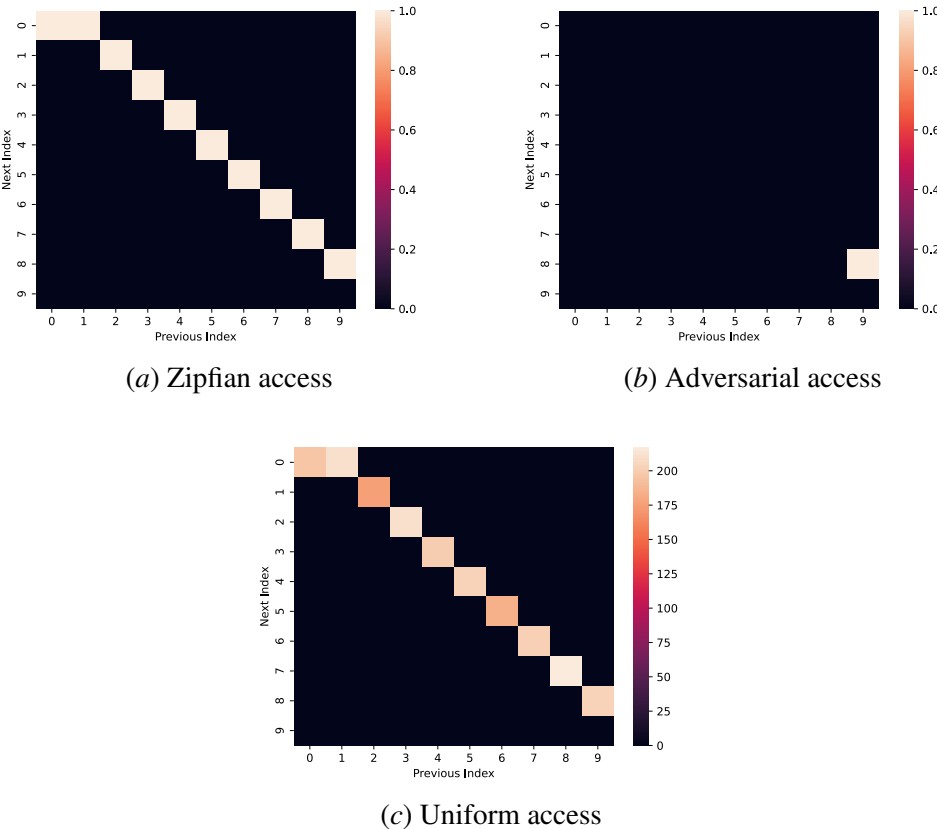

(a) Zipfian access

(b) Adversarial access

(c) Uniform access

Figure 5: Transposition Policy Map for list of size 10 on different access sequences.

8. We hypothesize that the learned policy is a **bucketing algorithm** which behaves as follows: it first creates buckets or subsections within the list and assigns items to each bucket based on their frequency which it learns over time. Items requested with a higher frequency are kept in buckets closer to the front of the list and items requested with a lower frequency are kept in buckets closer to the back of the list. Within each bucket, the learned policy maintains a Move-to-Front policy with little movement occurring between buckets especially as the list becomes stationary. To verify this hypothesis, we consider one additional piece of evidence: a transition graph for rearrangement choices during the testing phase. In Figure 9, we see a transition graph of the learned policy servicing requests from a Zipfian access sequence for a list of size 10. From the transition graph, we see that the two sub-sections or buckets in the list are [0,1,2,3] and [4,5,6,7,8,9]. Items are moved between the two buckets only twice. Almost all the rearrangements occur within the buckets. We also see that a lot of the movement within the buckets is towards the "front indices" which are [0,1] for the first bucket and [4,5] for the second bucket.

On one hand we can view this policy as a rough approximation of the frequency counter algorithm without explicitly maintaining counters. Here too as in Hsu et al. (2019), the deep Q-network learns which items are heavy i.e highly requested during the training phase. On the other hand, we can interpret this as some sort of interpolation between Move-to-Front and Transposition. Viewed as bucketing algorithms, Move-to-Front can be described as having a single bucket which is the whole

list and Transposition can be described as having $|L|$ buckets each of size 2. The learned policy, then, as an interpolation between the two algorithms selects the number of buckets to section the list into which ranges from 1 to $|L|$. The choice of number of buckets most likely depends on the access sequence it is servicing. In some sense, the learned policy represents a family of algorithms $\mathcal{F}$ parameterized by the number of buckets it chooses to section a list into. One area of possible is calculating the competitive ratio of algorithms in $\mathcal{F}$ which may be a function of the number of buckets chosen. The advantage of this perspective is that it treats Move-to-Front and Transposition as two variants of the same type of algorithm.

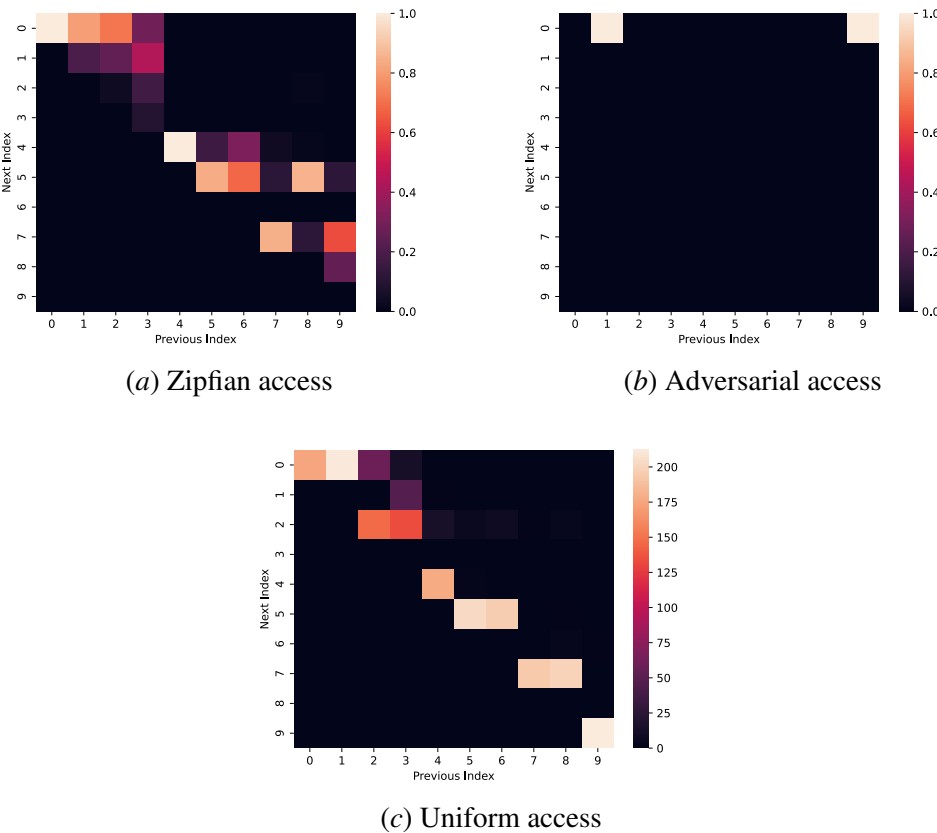

(*a*) Zipfian access            (*b*) Adversarial access

(*c*) Uniform access

Figure 6: Learned Policy Map for list of size 10 on different access sequences.

Finally, while we aimed to extract a policy and analyze its behavior, we recognized that a useful by-product of our work is a reinforcement learning agent that can adjust or modify its behavior to service any access sequence near-optimally. Going back to our interpolation interpretation above, this reinforcement learning agent is able to pick the right algorithm from $\mathcal{F}$ for a given access sequence. In fact, even when the support of the distribution or the kind of distribution is changed, the reinforcement learning agent is still able to select the right algorithm from $\mathcal{F}$. [8] Being able to dynamically adjust its behavior is an advantage the reinforcement learning agent has over existing

---

8. Results for this can be found in Appendix B.

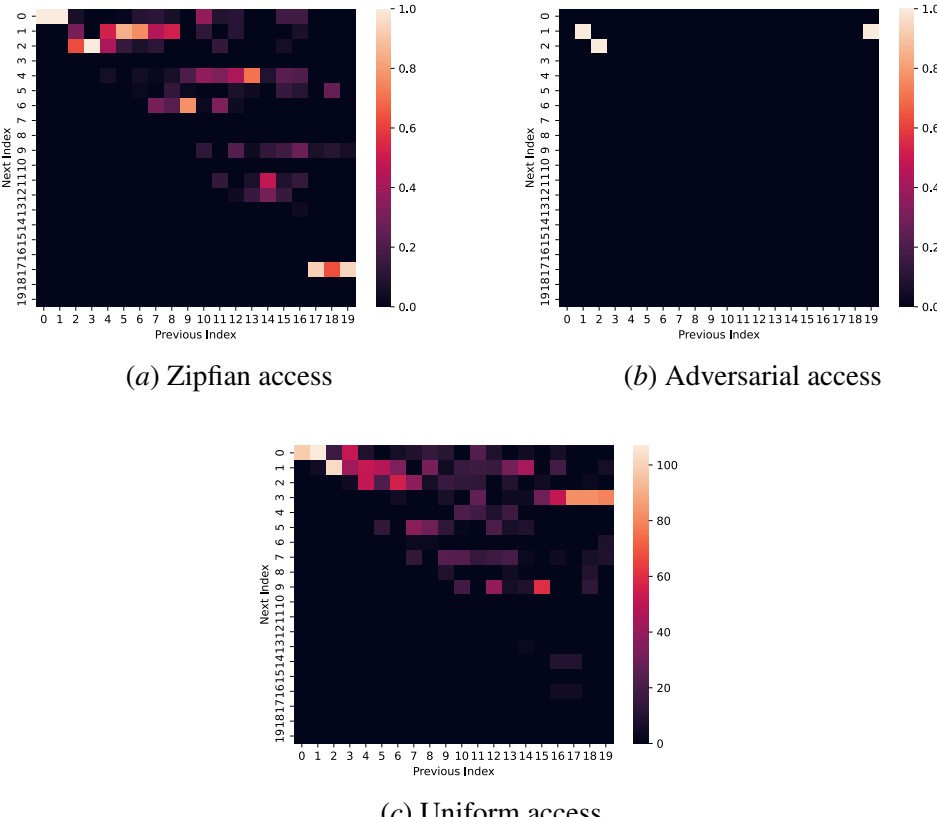

(a) Zipfian access       (b) Adversarial access

(c) Uniform access

Figure 7: Learned Policy Map for list of size 20 on different access sequences.

classical algorithms because once deployed with appropriate feedback signals, it can self-correct without any intervention.

## Acknowledgments

PI was supported in part by the NSF TRIPODS program (award DMS-2022448). RR was supported by the NSF TRIPODS program (award DMS-2022448) and CCF-2310818.

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

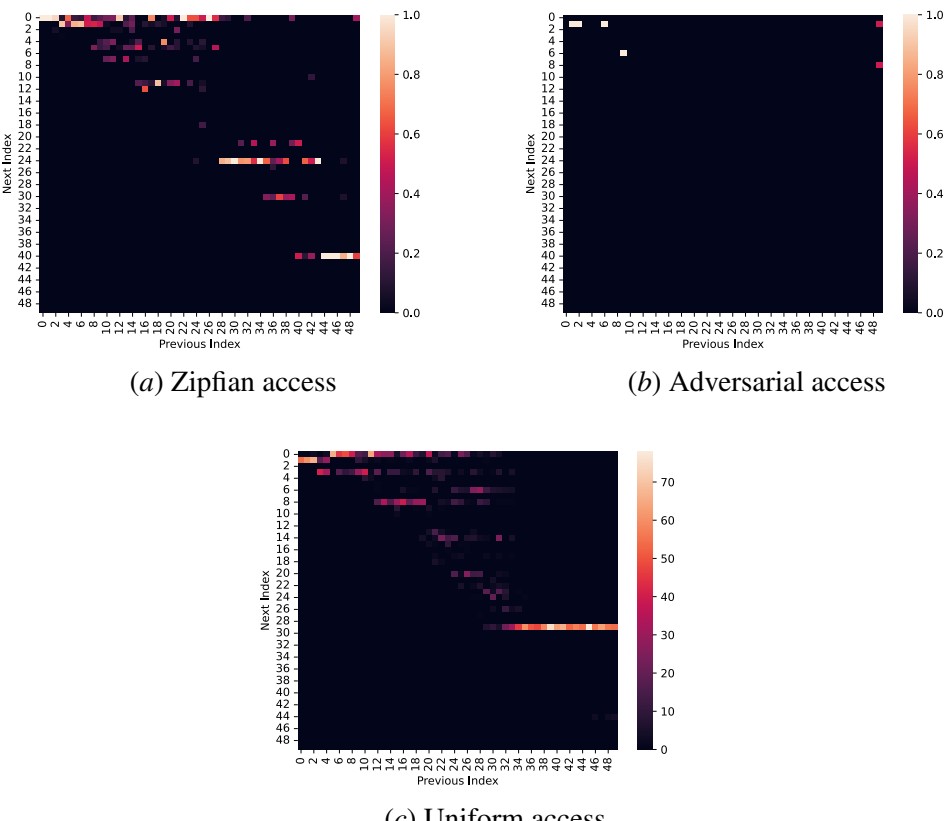

(a) Zipfian access

(b) Adversarial access

(c) Uniform access

Figure 8: Learned Policy Map for list of size 50 on different access sequences.

Ran Bachrach and Ran El-Yaniv. Online list accessing algorithms and their applications: Recent empirical evidence. In *Proceedings of the eighth annual ACM-SIAM symposium on Discrete algorithms*, pages 53–62, 1997.

Jon L Bentley, Kenneth L Clarkson, and David B Levine. Fast linear expected-time algorithms for computing maxima and convex hulls. *Algorithmica*, 9:168–183, 1993.

PJ Burville and JFC Kingman. On a model for storage and search. *Journal of Applied Probability*, 10(3):697–701, 1973.

David Gamarnik and Petar Momčilović. A transposition rule analysis based on a particle process. *Journal of applied probability*, 42(1):235–246, 2005.

WJ Hendricks. The stationary distribution of an interesting markov chain. *Journal of Applied Probability*, 9(1):231–233, 1972.

WJ Hendricks. An extension of a theorem concerning an interesting markov chain. *Journal of Applied Probability*, 10(4):886–890, 1973.

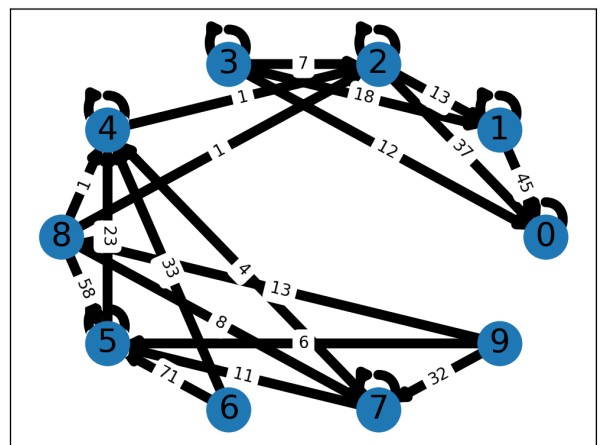

Figure 9: Transition Graph for Learned Algorithm under Zipfian distribution.

Chen-Yu Hsu, Piotr Indyk, Dina Katabi, and Ali Vakilian. Learning-based frequency estimation algorithms. In *International Conference on Learning Representations*, 2019.

Sandy Irani. Two results on the list update problem. *Information Processing Letters*, 38(6):301–306, 1991.

Sandy Irani, Nick Reingold, Jeffery Westbrook, and Daniel D Sleator. Randomized competitive algorithms for the list update problem. In *Proceedings of the second annual ACM-SIAM symposium on Discrete algorithms*, pages 251–260, 1991.

Yi C Kan and Sheldon M Ross. Optimal list order under partial memory constraints. *Journal of Applied Probability*, 17(4):1004–1015, 1980.

Donald Ervin Knuth. *The art of computer programming*, volume 3. Pearson Education, 1997.

John McCabe. On serial files with relocatable records. *Operations Research*, 13(4):609–618, 1965.

Volodymyr Mnih. Playing atari with deep reinforcement learning. *arXiv preprint arXiv:1312.5602*, 2013.

Nick Reingold, Jeffery Westbrook, and Daniel D Sleator. Randomized competitive algorithms for the list update problem. *Algorithmica*, 11(1):15–32, 1994.

Ronald Rivest. On self-organizing sequential search heuristics. *Communications of the ACM*, 19(2): 63–67, 1976.

Geza Schay, Jr and Francis W Dauer. A probabilistic model of a self-organizing file system. *SIAM Journal on Applied Mathematics*, 15(4):874–888, 1967.

Daniel D Sleator and Robert E Tarjan. Amortized efficiency of list update and paging rules. *Communications of the ACM*, 28(2):202–208, 1985.

Aaron M Tenenbaum and Richard M Nemes. Two spectra of self-organizing sequential search algorithms. *SIAM Journal on Computing*, 11(3):557–566, 1982.

Rahul Vaze. *Online Algorithms*. Cambridge University Press, 2023.

Wikipedia contributors. Rearrangement inequality — Wikipedia, the free encyclopedia, 2024. URL https://en.wikipedia.org/w/index.php?title=Rearrangement_inequality&oldid=1192950539. [Online; accessed 01-May-2024].

## Appendix A. Omitted Proofs

We recall the omitted lemma and theorem statements and give full proofs.

**Lemma 2**

$$\sum_{\pi \in \text{Case 1}} Q(\pi) \leq \frac{p_{j-2k}}{p_j} \cdot \sum_{\pi' \in \{f(\pi) | \pi \in \text{Case 1}\}} Q(\pi'). \tag{6}$$

**Proof** Note that we defined $\pi' = f(\pi)$ by swapping the element immediately to the left of $j$ with $j$. Note that $\pi' \in D_{t+1}$ and let $f(\pi) = \pi'$. By Lemma 1, we can compute that

$$Q(\pi') \geq \frac{p_j}{p_{j-2k}} \cdot Q(\pi).$$

Note that the image under $f$ for $\pi$ in Case 1 are all disjoint. Hence, the above equation implies the lemma statement. ∎

**Lemma 3**

$$\sum_{\pi \in \text{Case 2}} Q(\pi) \leq \sum_{\pi' \in \{f(\pi) | \pi \in \text{Case 2}\}} Q(\pi'). \tag{7}$$

*Furthermore, for any $\pi$ in Case 2, we have $f(\pi) \in D_m$ for $m > T$.*

**Proof** Let $e$ be the element directly to the left of $j$. Since we are in Case 2, it must be that $e < j - 2k$. Recall that for Case 2, we let $f(\pi)$ denote the permutation after performing the following two swaps: first swap $e$ with the left most element among $j - k, \ldots, j - 1$, then swap $j$ with the left most element among $j - k, \ldots, j - 1$ (in the current state of the list).

We can check that all the resulting images under $f$ are distinct. For example (in one subcase), we can write $\pi = B_1 e' B_2 e'' B_3 e j B_4$ where $B_1, \ldots, B_4$ represent some blocks of elements, $e', e'' \in \{j - k, \ldots, j - 1\}$ are the two left most elements in that range.

Then $f(\pi) = B_1 e B_2 j B_4 e' e'' B_4$. For two different permutations $\pi_1$ and $\pi_2$ in Case 2, either they have different tuples $(e, e', e'')$, or one of the blocks $B_i$ are different. The difference remains after applying $f$.

Now we again use Lemma to 1 to understand how the probabilities of the resulting lists (under $Q$) update. When swapping $e$ and $e'$, the probability (under $Q$) of the intermediate permutation

must grow by a factor at least $\left(\frac{p_e}{p_{e'}}\right)^{\ell}$, where $\ell$ is the number of distance we swapped. Then when swapping $e''$ and $j$, the probability can shrink, but this factor is not smaller than $\left(\frac{p_j}{p_{e''}}\right)^{\ell'}$, where $\ell'$ is the distance between $e''$ and $j$. Note that we must have $\ell' < \ell$ since we will never move $j$ before $e$ and furthermore, $e'/e > 1$ and $e''/j < 1$. Thus, denoting $\pi' = f(\pi)$, we have

$$Q(\pi') \geq Q(\pi) \cdot \left(\frac{e'}{e}\right)^{\alpha\ell} \cdot \left(\frac{e''}{j}\right)^{\alpha\ell'}.$$

Since $e' \cdot e'' \geq (j - k)^2 \geq j(j - 2k) > je$, we have $Q(\pi') > Q(\pi)$, as desired. Note that in $\pi'$, $j$ is still before $i$, but they are now separated by at least $k/2 - 1 > T$ elements since we assumed at least half of the indices in $\{j - k, \ldots, j - 1\}$ were to the left of $j$. $\blacksquare$

**Lemma 4**

$$\sum_{\pi \in Case\ 3} Q(\pi) \leq \left(1 - \frac{k}{3j}\right)^{\alpha k/2} \cdot \sum_{\pi' \in \{f(\pi)|\pi \in Case\ 3\}} Q(\pi'). \tag{8}$$

**Proof** Recall that for Case 3, we first swap $i$ with $j$. Then we swap the element $e$ among $j - k, \ldots, j - k/3$ that is in the *right most* position with $j$. Let $B_1jB_2iB_3eB_4$ represent $\pi$. Then $f(\pi) = B_1iB_2eB_3jB_4$. Thus for different permutations in Case 3, either they have different elements $e$ or one of the blocks $B_i$ are different, which also remains the case after applying $f$. We claim that using Lemma 1, the following holds for any $\pi' = f(\pi)$.

$$Q(\pi') \geq \left(\frac{j}{i}\right)^{\alpha t} \cdot \left(\frac{j}{j - k/3}\right)^{\alpha(k/2 - t)} \cdot Q(\pi).$$

The first factor comes from the fact that $j$ and $i$ are separated by $t$ elements. The second factor arises since we swapped $e$ with $j$ and they are separated by at least $k/2 - t$ positions (the right most element of $k/2$ elements, except the ones that might be between $i$ and $j$). This immediately implies the lemma as $j/i \geq j/(j - k/3)$. $\blacksquare$

**Theorem 3.1** *Let $b(i, j)$ be the probability that $i$ appears before $j$ under $Q$ and analogously define $b(j, i)$. Under the parameter settings of Section 3.2, we have*

$$\frac{b(i, j)}{b(j, i)} \geq (1 - o(1)) \frac{1}{4T} \cdot \left(\frac{j}{i}\right)^{\alpha T}.$$

**Proof** Note that

$$\frac{b(i, j)}{b(j, i)} = \frac{\sum_{\pi,\ i\ \text{appears before}\ j} Q(\pi)}{\sum_{\pi,\ j\ \text{appears before}\ i} Q(\pi)}.$$

Let $N_m$ denote the permutations in the numerator where $i$ and $j$ are separated by $m \geq 0$ elements and define $D_m$ to be the permutations in the denominator where they are separated by $m \geq 0$ elements. There is a natural one-to-one mapping between them by swapping the positions of $i$ and $j$. Now

decompose $D_m = D_m^1 \cup D_m^2 \cup D_m^3$ based on the three cases defined in Section 3.2 for $0 \le m \le T$. We know from Lemma 2 that

$$Q(D_m^1) \le \beta Q(D_{m+1}). \tag{9}$$

where $\beta = \left(1 + \frac{4k}{j}\right)^\alpha$. Furthermore, Lemma 3 implies

$$Q(D_m^2) \le \sum_{m'>T} Q(D_{m'}). \tag{10}$$

Our goal is to 'charge' the probabilities of the permutations $Q(D_m)$ for $0 \le m \le T$ to $Q(D_{m'})$ for $m' > T$. Towards this end, we iterate on Inequality 9. For any $0 \le m \le T$, we have

$$\begin{aligned} Q(D_m^1) &\le \beta Q(D_{m+1}) \\ &= \beta \left(Q(D_{m+1}^1) + Q(D_{m+1}^2) + Q(D_{m+1}^3)\right) \\ &\le \beta^2 Q(D_{m+2}) + \beta \left(Q(D_{m+1}^2) + Q(D_{m+1}^3)\right) \\ &\cdots \\ &\le \beta^{T+1} Q(D_{T+1}) + \beta^T \left(\sum_{s=0}^T Q(D_s^2) + Q(D_s^3)\right). \end{aligned}$$

Thus, using Inequality 10, we have

$$\sum_{s=0}^T Q(D_s^1) \le 2T\beta^{T+1} \sum_{m'>T} Q(D_{m'}) + \beta^T \sum_{s=0}^T Q(D_s^3).$$

Now we control the $Q(D_s^3)$ terms. From Lemma 4, we know that that for every $0 \le m \le T$, we have

$$Q(D_m^3) \le e^{-\alpha k^2/(6j)} Q(N_{m'}) \tag{11}$$

for some $m' \ge m + k/2 \gg T$. Thus,

$$\sum_{s=0}^T Q(D_s^1) \le 2T\beta^{T+1} \sum_{m'>T} Q(D_{m'}) + T\beta^T e^{-\alpha k^2/(6j)} \sum_{m'>T} Q(N_{m'}).$$

Note that we can simplify

$$T^2 \beta^T e^{-\alpha k^2/(6j)} \le T e^{-\alpha k^2/(6j) + 4kT\alpha/j + 2\log(T)} \le e^{-\alpha k^2/(10j)}$$

for our range of parameters from Section 3.2. Finally, we can now bound the ratio $\frac{b(i,j)}{b(j,i)}$. We have

$$\frac{\sum_s Q(N_s)}{\sum_s Q(D_s)} = \frac{\sum_s Q(N_s)}{\sum_{m=0}^T (Q(D_m^1) + Q(D_m^1) + Q(D_m^3)) + \sum_{m>T} Q(D_m)}.$$

Combining all of our inequalities means that the first term in the denominator can be bounded by

$$\sum_{m=0}^T (Q(D_m^1) + Q(D_m^1) + Q(D_m^3)) \le 3T\beta^{T+1} \sum_{m'>T} Q(D_{m'}) + e^{-\alpha k^2/(10j)} \sum_{m'>T} Q(N_{m'}),$$

resulting in

$$\frac{\sum_s Q(N_s)}{\sum_s Q(D_s)} \geq \frac{\sum_s Q(N_s)}{4T\beta^{T+1}\sum_{m'>T}Q(D_{m'}) + e^{-\alpha k^2/(10j)}\sum_{m'>T}^T Q(N_{m'})}$$

$$\geq \frac{2\sum_{m'>T}Q(N_{m'})/2}{4T\beta^{T+1}\sum_{m'>T}Q(D_{m'}) + e^{-\alpha k^2/(10j)}\sum_{m'>T}^T Q(N_{m'})}$$

$$\geq \frac{\left(\frac{p_i}{p_j}\right)^{T+1}\sum_{m'>T}Q(D_{m'}) + \frac{1}{2}\sum_{m'>T}Q(N_{m'})}{4T\beta^{T+1}\sum_{m'>T}Q(D_{m'}) + e^{-\alpha k^2/(10j)}\sum_{m'>T}^T Q(N_{m'})},$$

where we have used the fact that $Q(N_s) \geq \left(\frac{p_i}{p_j}\right)^n \cdot Q(D_s)$ for all $n$ due to Lemma 1. Finally, using the fact that $(a+b)/(c+d) \geq \min(a/c, b/d)$ for positive real numbers $a, \dots, d$, we have

$$\frac{b(i,j)}{b(j,i)} = \frac{\sum_s Q(N_s)}{\sum_s Q(D_s)} \geq \min\left(\frac{1}{4T} \cdot \left(\frac{j}{i}\right)^{\alpha(T+1)} \cdot \frac{1}{(1+4k/j)^{T+1}}, \frac{e^{\alpha k^2/(10j)}}{2}\right)$$

$$\geq (1-o(1))\frac{1}{4T} \cdot \left(\frac{j}{i}\right)^{\alpha T}.$$

$\blacksquare$

### A.1. Proving the Main Theorem

In this section, we bound the three estimates from Section 3.3 and prove Theorem 1.1. First we give prove some useful auxiliary results.

Let $C_\alpha = \sum_{i=1}^n \frac{1}{i^\alpha}$ (we hide the dependence on $n$). The optimum cost is given by

$$\text{OPT}(\alpha) = \frac{1}{C_\alpha}\sum_{i=1}^n \frac{1}{i^\alpha} \cdot i = \frac{C_{\alpha-1}}{C_\alpha}.$$

It is easy to see that in this range, $\text{OPT}(\alpha) = \omega(1)$.

**Lemma 5** *For $0 < \alpha \leq 2$,*

$$\lim_{n\to\infty} \frac{C_{\alpha-1+0.1}}{C_{\alpha-1}} = 0.$$

**Proof** We briefly sketch the proof. Let $k = \alpha - 1$. When $-1 < k \leq 0$, the exponent of $C_k$ is negative so all the terms in the sum are integers, meaning the sum is increasing. Furthermore, every term of $C_k$ is larger than the corresponding term of $C_{k+0.1}$ and after the $n/2$-th term, the values are larger by a factor of $\Theta(n^{0.1})$.

For $0 < k < 1$, we can lower bound

$$C_k \geq \int_1^n \frac{1}{x}\,dx = \Theta(n^{1-k})$$

and on the other hand, either $C_{k+0.1}$ is $O(\log n)$, if $k + 0.1 \geq 1$, or it can be upper bounded by

$$C_{k+0.1} \leq \int_0^n \frac{1}{x}\,dx = \Theta(n^{0.9-k}),$$

which is asymptotically smaller.

For $k = 0$, the numerator is constant and the denominator is $\Theta(\log n)$, which covers all the cases.

∎

Now we bound the three estimates.

**Lemma 6 (Estimate 1)** *If $p_i \propto 1/i^\alpha$ for $0 < \alpha \leq 2$, then*

$$\sum_{i=1}^{n} p_i \left( \sum_{1 \leq j \leq i+i^{0.9}} b(j,i) \right) = (1 + o(1)) \cdot \text{OPT}(\alpha).$$

**Proof** Note that $\text{OPT}(\alpha) = \frac{C_{\alpha-1}}{C_\alpha}$. It is proven in Rivest (1976) that for every $i < j$ (so that $p_i > p_j$), we have $b(i,j) \geq \frac{p_i}{p_i+p_j}$, see Section 1) . Since $b(j,i) + b(i,j) = 1$, this implies that $b(j,i) \leq \frac{p_j}{p_i+p_j}$ for $i < j$. Note the upper bound $b(i,j) \leq 1$ always holds for all $i,j$. We have

$$\sum_i p_i \cdot \left( \sum_{1 \leq j \leq i+i^{0.9}} b(j,i) \right) \leq \frac{1}{C_\alpha} \sum_i \frac{1}{i^\alpha} \cdot \left( i + \sum_{i < j \leq i+i^{0.9}} b(j,i) \right)$$

$$\leq \frac{C_{\alpha-1}}{C_\alpha} + \frac{1}{C_\alpha} \sum_i \frac{1}{i^\alpha} \sum_{i < j \leq i+i^{0.9}} \frac{p_j}{p_i + p_j}$$

$$= \frac{C_{\alpha-1}}{C_\alpha} + \frac{1}{C_\alpha} \sum_i \sum_{i < j \leq i+i^{0.9}} \frac{1}{i^\alpha + j^\alpha}$$

$$\leq \frac{C_{\alpha-1}}{C_\alpha} + \frac{1}{C_\alpha} \sum_i \int_{2i}^{2i+i^{0.9}} \frac{1}{x^\alpha} \, dx.$$

Now for $\alpha = 1$, we have

$$\sum_i \int_{2i}^{2i+i^{0.9}} \frac{1}{x^\alpha} \, dx = \sum_i \log \left( \frac{1}{2i^{0.1}} + 1 \right) = o(n),$$

so we have

$$\sum_i p_i \cdot \left( \sum_{1 \leq j \leq i+i^{0.9}} b(j,i) \right) \leq \frac{n}{H_n} + \frac{1}{H_n} \cdot o(n) = (1 + o(1))\text{OPT}(\alpha),$$

in the $\alpha = 1$ case, where $H_n$ denotes the $n$th harmonic number. For $0 < \alpha < 1$, we have

$$\sum_i \int_{2i}^{2i+i^{0.9}} \frac{1}{x^\alpha} \, dx = \sum_i \frac{(2i)^{1-\alpha}}{1 - \alpha} \left( \left( 1 + \frac{1}{2i^{0.1}} \right)^{1-\alpha} - 1 \right)$$

$$\leq \sum_i \frac{(2i)^{1-\alpha}}{1 - \alpha} \cdot \frac{1 - \alpha}{2i^{0.1}}$$

$$= \frac{1}{2^\alpha} \cdot \sum_i i^{0.9-\alpha}$$

$$= o(C_{\alpha-1}),$$

where the first inequality is Bernoulli's inequality and the last step follows from Lemma 5. Using the fact that $\text{OPT}(\alpha) = \omega(1)$, we again have that

$$\sum_i p_i \cdot \left( \sum_{1 \leq j \leq i+i^{0.9}} b(j,i) \right) \leq (1 + o(1))\text{OPT}(\alpha)$$

in this case. Finally in the case $\alpha > 1$,

$$\sum_i \int_{2i}^{2i+i^{0.9}} \frac{1}{x^\alpha}\, dx = \sum_i \frac{(2i)^{1-\alpha}}{\alpha - 1} \left( 1 - \left( 1 + \frac{1}{2i^{0.1}} \right)^{1-\alpha} \right)$$

and Bernoulli's inequality again implies that

$$\left( 1 + \frac{1}{2i^{0.1}} \right)^{1-\alpha} \geq 1 + \frac{1 - \alpha}{2i^{0.1}}$$

and the same steps above also imply that the total sum is bounded by $(1 + o(1))\text{OPT}(\alpha)$. ∎

**Lemma 7 (Estimate 2)**  *If $p_i \propto 1/i^\alpha$ for $0 < \alpha \leq 2$, then*

$$\sum_{i \leq \log n} p_i \left( \sum_{j > i+i^{0.9}} b(j,i) \right) = o(\text{OPT}(\alpha)).$$

**Proof** First we consider the $\alpha > 1$ case. We have

$$\begin{aligned}
\sum_{i \leq \log n} p_i \cdot \left( \sum_{j > i+i^{0.9}} b(j,i) \right) &\leq \frac{1}{C_\alpha} \sum_{i \leq \log n} \sum_{j > i+i^{0.9}} \frac{1}{i^\alpha + j^\alpha} \\
&\leq \frac{1}{C_\alpha} \sum_{i \leq \log n} \sum_{j \geq i+1} \frac{1}{j^\alpha} \\
&\leq \frac{1}{C_\alpha} \sum_{i \leq \log n} \int_i^\infty \frac{1}{x^\alpha}\, dx \\
&= \frac{1}{(\alpha - 1)C_\alpha} \sum_{i \leq \log n} \frac{1}{i^{\alpha-1}}.
\end{aligned}$$

For $2 > \alpha > 1$, we have $\sum_{i \leq \log n} \frac{1}{i^{\alpha-1}} = \Theta((\log n)^{2-\alpha})$, but $C_{\alpha-1} = \sum_{1 \leq i \leq n} \frac{1}{i^{\alpha-1}} = \Theta(n^{2-a})$. For $\alpha = 2$, we have $\sum_{i \leq \log n} \frac{1}{i^{\alpha-1}} = \Theta(\log \log n)$ but $C_{\alpha-1} = \Theta(\log n)$, so in both cases, we have $\sum_{i \leq \log n} \frac{1}{i^{\alpha-1}} = o(C_{\alpha-1})$. In other words, we have

$$\sum_{i \leq \log n} p_i \cdot \left( \sum_{j > i+i^{0.9}} b(j,i) \right) = o\left(\text{OPT}(\alpha)\right).$$

For $\alpha = 1$, we have

$$\sum_{i \leq \log n} p_i \cdot \left( \sum_{j > i + i^{0.9}} b(j,i) \right) \leq \frac{1}{H_n} \sum_{i \leq \log n} \sum_{j > i + i^{0.9}} \frac{1}{i+j} = O(\log n),$$

so again the sum is $o(\text{OPT}(1))$ as $\text{OPT}(1) = \Omega(n/\log n)$. We now consider $0 \leq \alpha < 1$. We have

$$\sum_{i \leq \log n} \sum_{j \geq i+1} \frac{1}{j^\alpha} = O((\log n)n^{1-\alpha})$$

and $C_\alpha = \sum_{1 \leq i \leq n} 1/i^\alpha = \Theta(n^{1-\alpha})$, so

$$\sum_{i \leq \log n} p_i \cdot \left( \sum_{j > i + i^{0.9}} b(j,i) \right) = O(\log n),$$

but $\text{OPT}(\alpha) = \Omega(n)$ in this case, implying that in this case, it also holds that

$$\sum_{i \leq \log n} p_i \cdot \left( \sum_{j > i + i^{0.9}} b(j,i) \right) = o(\text{OPT}(\alpha)),$$

as desired. ∎

**Lemma 8 (Estimate 3)** *If $p_i \propto 1/i^\alpha$ for $0 < \alpha \leq 2$, then*

$$\sum_{i > \log n} p_i \left( \sum_{j > i + i^{0.9}} b(j,i) \right) = o(\text{OPT}(\alpha)).$$

**Proof** In this case, Theorem 3.1 states

$$\frac{b(i,j)}{b(j,i)} \geq \frac{1}{8T} \cdot \left( \frac{j}{i} \right)^{\alpha T} \implies b(i,j) \geq \frac{j^{\alpha T}}{8T \cdot i^{\alpha T} + j^{\alpha T}},$$

where recall that $T = i^{0.15}$. Now we bound the sum as

$$\frac{1}{C_\alpha} \sum_{i > \log n} \frac{1}{i^\alpha} \sum_{j > i + i^{0.9}} b(j, i) \leq \frac{1}{C_\alpha} \sum_{i=1}^{n} \sum_{j > i + i^{0.9}} \frac{8T \cdot i^{\alpha(T-1)}}{8T \cdot i^{\alpha(T-1)} + j^{\alpha T}}$$

$$= \frac{1}{C_\alpha} \sum_{i=1}^{n} \sum_{j > i + i^{0.9}} \frac{1}{1 + j^{\alpha T}/(8T i^{\alpha(T-1)})}$$

$$\leq \frac{1}{C_\alpha} \sum_{i=1}^{n} 8T i^{\alpha(T-1)} \int_{i + i^{0.9}}^{\infty} \frac{1}{x^{\alpha T}} \, dx$$

$$\leq \frac{1}{C_\alpha} \sum_{i=1}^{n} \frac{8T i^{\alpha(T-1)}}{(i + i^{0.9})^{\alpha(T-1)}(T - 1)}$$

$$= \frac{1}{C_\alpha} \cdot \sum_{i=1}^{n} \frac{8T}{T - 1} \cdot \left( 1 - \frac{1}{1 + i^{0.1}} \right)^{\alpha(T-1)}$$

$$\leq \frac{1}{C_\alpha} \cdot \sum_{i=1}^{n} \frac{8T}{T - 1} \cdot e^{-\frac{\alpha(T-1)}{1 + i^{0.1}}}$$

$$= \frac{1}{C_\alpha} \cdot \sum_{i=1}^{n} e^{-\Theta(\alpha \cdot i^{0.05})}$$

$$= o\left( \text{OPT}(\alpha) \right),$$

where we have used the fact that $C_{\alpha-1} = \omega(1)$ and $\sum_{i=1}^{n} e^{-\Theta(\alpha \cdot i^{0.05})} = O(1)$. ∎

Putting together the above lemmas proves the main theorem.

**Proof** [Proof of Theorem 1.1] From Lemmas 6, 7, and 8, we have

$$\text{Cost}(\text{Transposition}, \text{Zipfian}(\alpha)) = \sum_{i=1}^{n} p_i \cdot \left( 1 + \sum_{1 \leq j \leq n, j \neq i} b(j, i) \right)$$

$$= 1 + \underbrace{\sum_{i=1}^{n} p_i \left( \sum_{1 \leq j \leq i + i^{0.9}} b(j, i) \right)}_{\text{Estimate 1}} + \underbrace{\sum_{i \leq \log n} p_i \left( \sum_{j > i + i^{0.9}} b(j, i) \right)}_{\text{Estimate 2}} + \underbrace{\sum_{i > \log n} p_i \left( \sum_{j > i + i^{0.9}} b(j, i) \right)}_{\text{Estimate 3}}$$

$$\leq (1 + o(1))\text{OPT}(\alpha) + o(\text{OPT}(\alpha)) + o(\text{OPT}(\alpha))$$

$$= (1 + o(1))\text{OPT}(\alpha),$$

as desired. ∎

## Appendix B. Omitted Experimental Results and Discussions

### B.1. Evaluating the reinforcement learning agent

As mentioned in Section 5, we perform additional experiments to evaluate the performance of the reinforcement learning agent when there is a change in the distribution $\mathcal{D}$, the list $L$ or both. The time

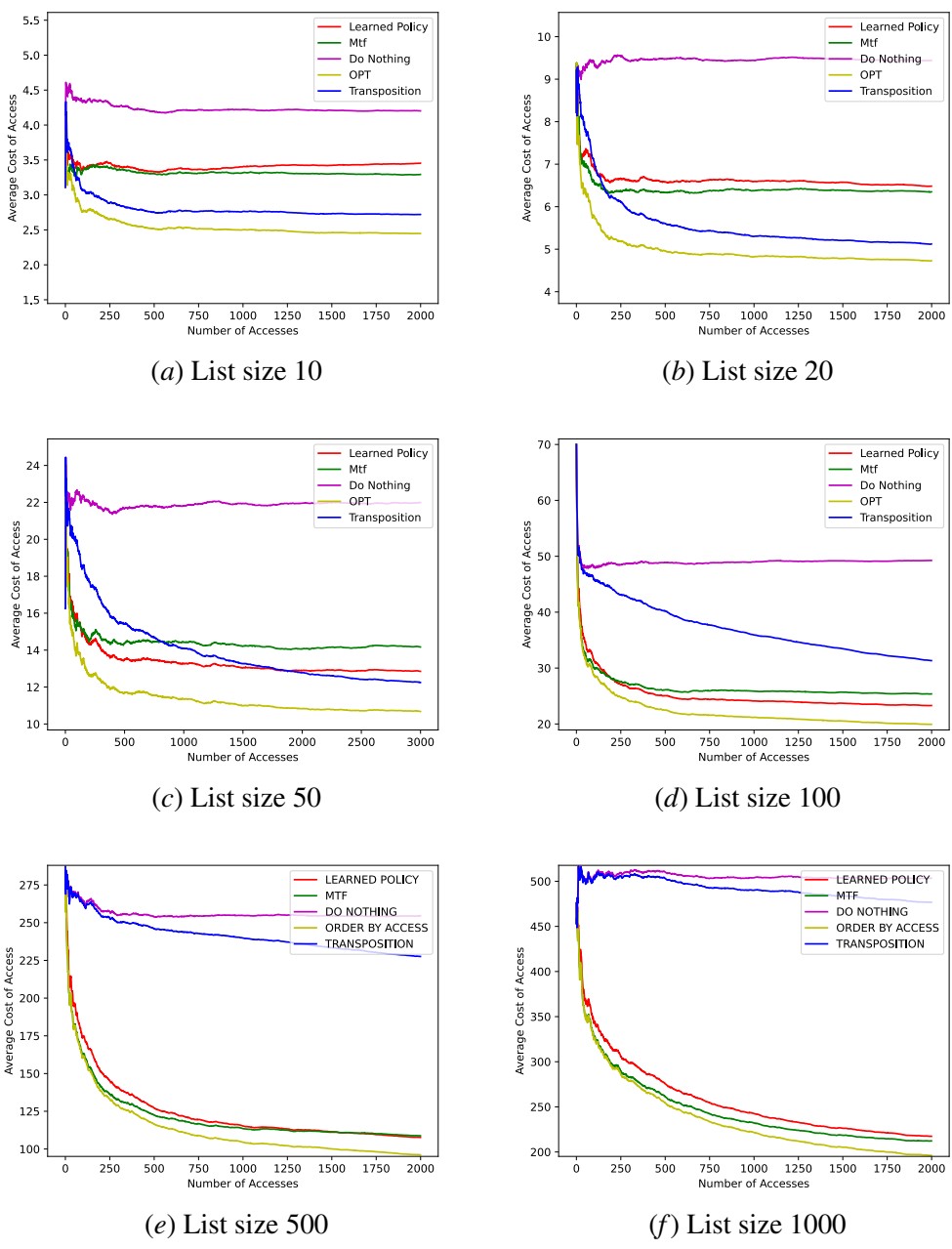

(*a*) List size 10

(*b*) List size 20

(*c*) List size 50

(*d*) List size 100

(*e*) List size 500

(*f*) List size 1000

Figure 10: Access cost for Zipfian distribution on lists of various sizes

step where this occurs marks the end of what we call an **episode** and the beginning of another. Each episode lasts 5000 timesteps and an experiment has 10 episodes. The kinds of episode-to-episode variations we consider are:

1. **No list change, Distribution change**: From episode to episode, the list $L$ remains unchanged and hence the support of the distribution remains unchanged. The only thing that changes is the

distribution frequency over the keys. So, for instance, the most frequently accessed item changes from episode to episode but the list is itself is unchanged. This experiment exists to ascertain the adaptability of the learned algorithm. If it can recognize that the most frequently accessed items have changed and modify its behavior accordingly, then it has achieved adaptability.

2. **List change, No Distribution change**: Here, rather than change the access frequency distribution, we keep the distribution the same and change the support i.e. the list $L$. Again, this is a test of adaptability. We want to ensure that the learned algorithm does not merely memorize specific key values which it carries from episode to episode.

3. **List change, Distribution change**: Here, both the list and the distribution changes. This is also a test of adaptability, to verify that the learned algorithm can adjust its policy in response to workload changes.

4. **List change & No list change**: For the adversarial generation, we have two variants. In one variant, we change the list from episode to episode and in another variant, we do not change the underlying list $L$.

We describe the modified experimental steps below:

1. Each experiment begins with choosing a list size $|L|$ from $[10, 20, 50, 100, 500, 1000]$ and choosing $|L|$ unique integers uniformly from the range 1 and 100000. We also choose which type of episode-to-episode variation we would like to have. As we saw above, the choice of variations available depends on the distribution ($\mathcal{D}$) we choose for the experiment.

2. Once we have our list of integers, we sample 10 request sequences each of length 5000 from $\mathcal{D}$ which we call $R_1, R_2, \cdots, R_{10}$ as dictated by our choice of episode-to-episode variation.

3. For the next 50000 timesteps, agent receives an observation from the environment in the form $(r_t, L(r_t))$, formats the state from the observation appropriately and uses the deep neural network to estimate the best action to take. Experience replay and transferring of weight to the target network happen periodically. In the first 5000 timesteps, $r_t$ comes from $R_1$, in the second 5000 timesteps, $r_t$ comes from $R_2$ and so and so forth.

4. In addition to the agent's list, we create 4 other lists for four other list update policies we will be using as benchmarks: Move-To-Front, Transposition, a policy that never moves any item(Do Nothing) and a policy we call OPT that keeps frequency counters and orders items by their frequency. Each policy receives the same access request $r_t$ and maintains their list according to their respective policy. This means we have 5 different lists for each of the 5 algorithms we are evaluating: $L_{agent}, L_{MTF}, L_{Trans}, L_{Do-Nothing}, L_{OPT}$

5. During each episode, we record the move choices of each policy and the average cost per access over the 3000 time steps.

6. We do not reset the network between episodes. In fact from the perspective of the agent, it knows nothing as changed until it learns or detects the change on its own and acts accordingly.

7. We repeat the above procedure 10 times and each time we record move choices and the average cost per access.

The desired behaviour is for the learned algorithm's performance to remain competitive across episode-to-episode variations. In Figures 11(*c*), 11(*b*), 11(*d*) and 11(*a*) we see that indeed the learned algorithm maintains it's competitive performance from episode to episode.

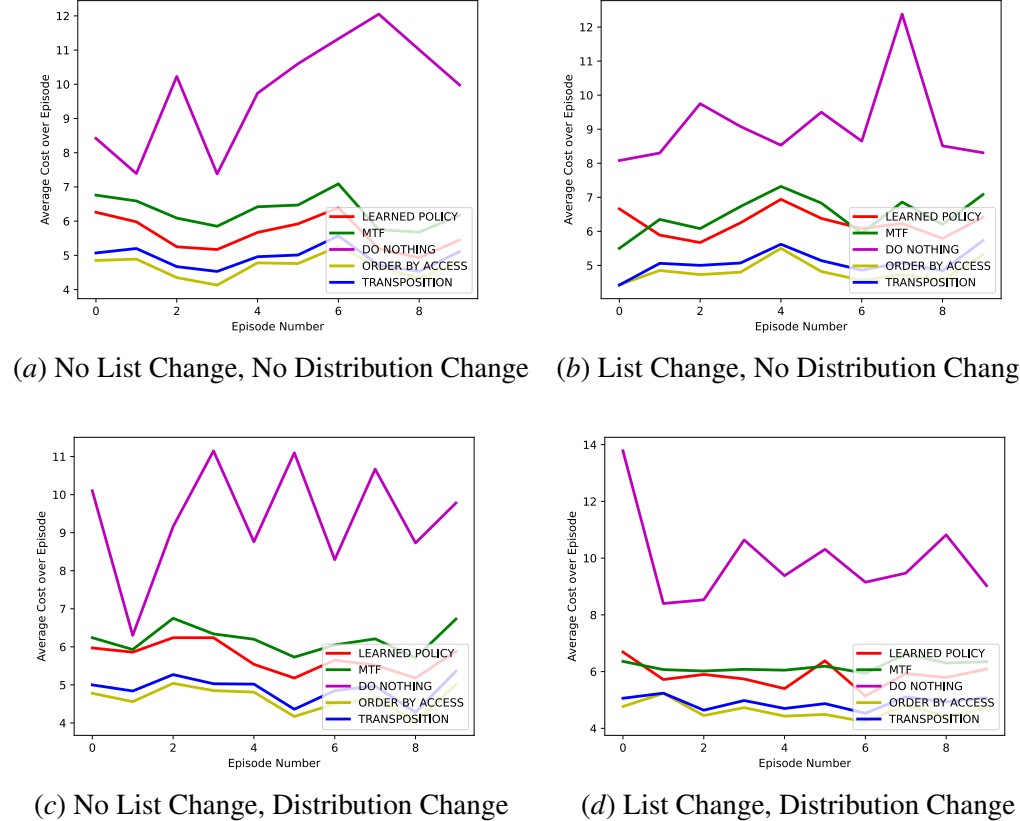

(*a*) No List Change, No Distribution Change    (*b*) List Change, No Distribution Change

(*c*) No List Change, Distribution Change    (*d*) List Change, Distribution Change

Figure 11: Access cost for Zipfian distribution on lists of various sizes

## B.2. Factors affecting bucketing in learned policy

1. Length of training period: Longer training periods allow the agent to better learn the frequency distribution over integers resulting in more fine-grained bucketing.

2. Size of Replay Buffer: A large replay buffer means the agent has more samples from $\mathcal{D}$ and can learn the frequency distribution better resulting in more fine-grained bucketing.

3. Set of Action Space: Limiting the action space to $[0 \cdots L(r_t)]$ can force the agent to limit the number of buckets.

4. Choice of state: Different choices of succinct state affects the quality of the Q-function learned which in turn affects the agent's ability to learn $D$ affecting the quality of bucketing.

### B.3. Choosing the right transformation function ($C$)

We considered many candidates for $C$. We describe each one below:

1. Permutation Matrix: This list representation is much like a standard permutation matrix consisting of only 0s and 1s with each row and column having only a single entry being equal to 1. For a list $L$, the row of the permutation matrix represents the position of a record in the initial permutation of $L$ and the column represents the position it has been moved to. This list state representation has size $|L|^2$ and the total size of the state representation is $|L|^2 + 1$. The size of the total state space is $|L| \cdot 2^{|L|}$.

2. First Item: The first record in the list $L$. This list state representation has size 1 and the total size of the state representation is 2. The size of the total state space is $|L|^2$.

3. Preceding Item: Given that $q_t$ references the record in position $i$, the preceding item list state representation chooses the record $i - 1$ as the list state. If $i = 1$, then the preceding item list state chooses the record $i$ itself. This list state representation has size 1 and the total size of the state representation is 2. The size of the total state space is $|L|^2$.

4. Succeeding Item: Given that $q_t$ queries the record in position $i$, the succeeding item list state representation chooses the record $i + 1$ as the list state. If $i = |L|$, then the succeeding item list state chooses the record $i$ itself. This list state representation has size 1 and the total size of the state representation is 2. The size of the total state space is $|L|^2$.

5. Preceding and Succeeding Item: This list state representation uses both the preceding and succeeding item representation. This list state representation has size 2 and the total size of the state representation is 3. The size of the total state space is $\leq |L|^3$.

6. Current Position: Given that $q_t$ queries the record in position $i$, the current position list state representation chooses the index $i$ as the list state. This list state representation has size 1 and the total size of the state representation is 2. The size of the total state space is $|L|^2$.

7. First $\log$ N: The first $\log |L|$ records of $L$ are the list state representation. This list state representation has size $\log |L|$ and the total size of the state representation is $\log |L| + 1$. The size of the total state space is $|L| \cdot \binom{|L|}{\log |L|}$.

8. Last $\log$ N: The last $\log |L|$ records of $L$ are the list state representation. This list state representation has size $\log |L|$ and the total size of the state representation is $\log |L| + 1$. The size of the total state space is $|L| \cdot \binom{|L|}{\log |L|}$.

9. First member of $\log$ N: Given a list $L$, break it up into $\frac{|L|}{\log |L|}$ sections each of size $\log |L|$. The list state representation consists of the first record of each $\log |L|$ chunk. This list state representation has size $\frac{|L|}{\log |L|}$ and the total size of the state representation is $\frac{|L|}{\log |L|} + 1$. The size of the total state space is $|L| \cdot \binom{|L|}{\frac{|L|}{\log |L|}}$.

To choose between these different transformation functions, we look at the average cost of the learned algorithm when we use each of these transformation functions. Below we show the result for a Zipf(1) access sequence:

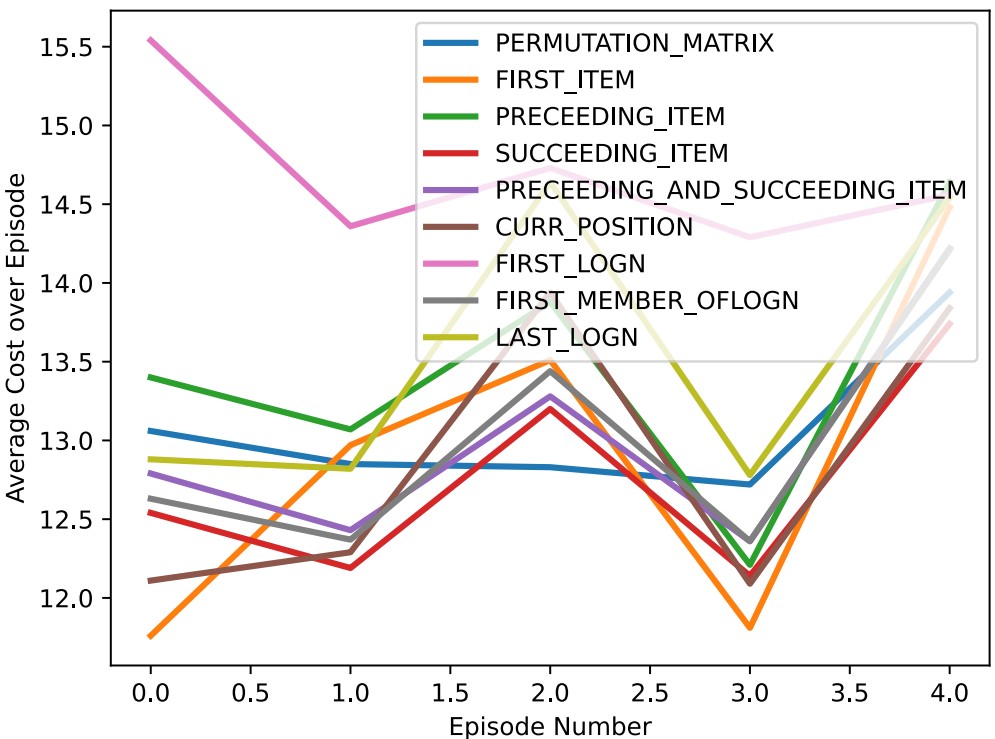

Figure 12: Average cost of learned algorithm on Zipfian query sequence on a list of size 50 for different state representations. List remains unchanged and Distribution remains unchanged from episode to episode.

## B.4. Ommited Results and Graphs

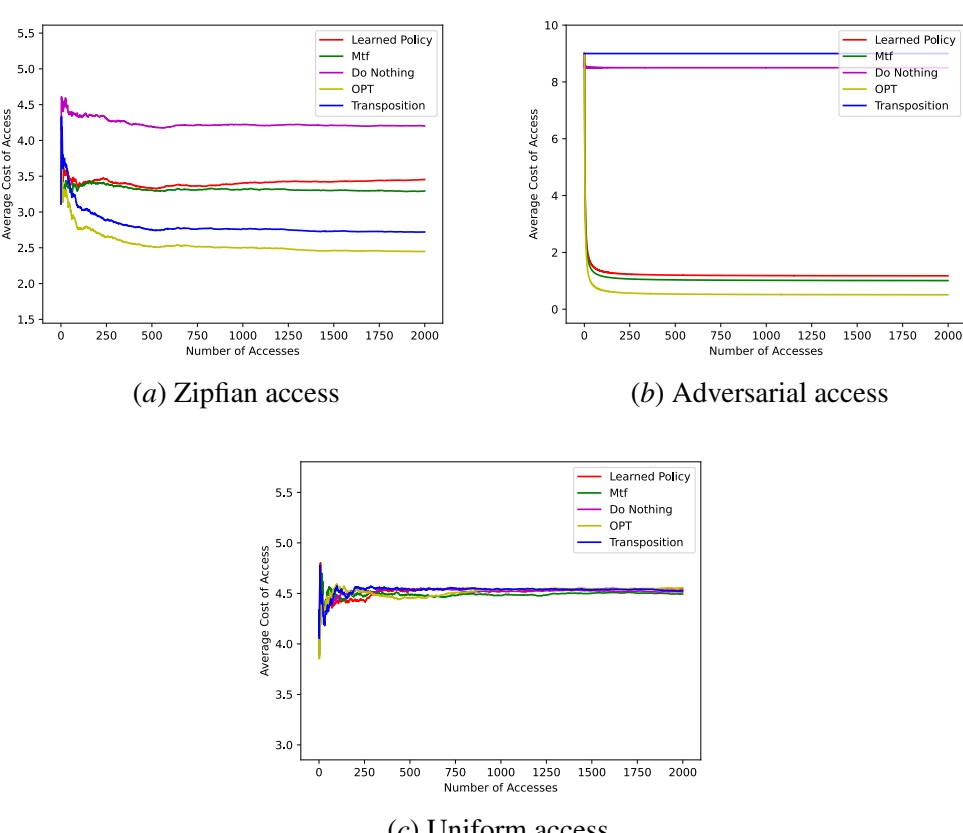

(*a*) Zipfian access

(*b*) Adversarial access

(*c*) Uniform access

Figure 13: Average cost for a list of size 10 on different access sequences.

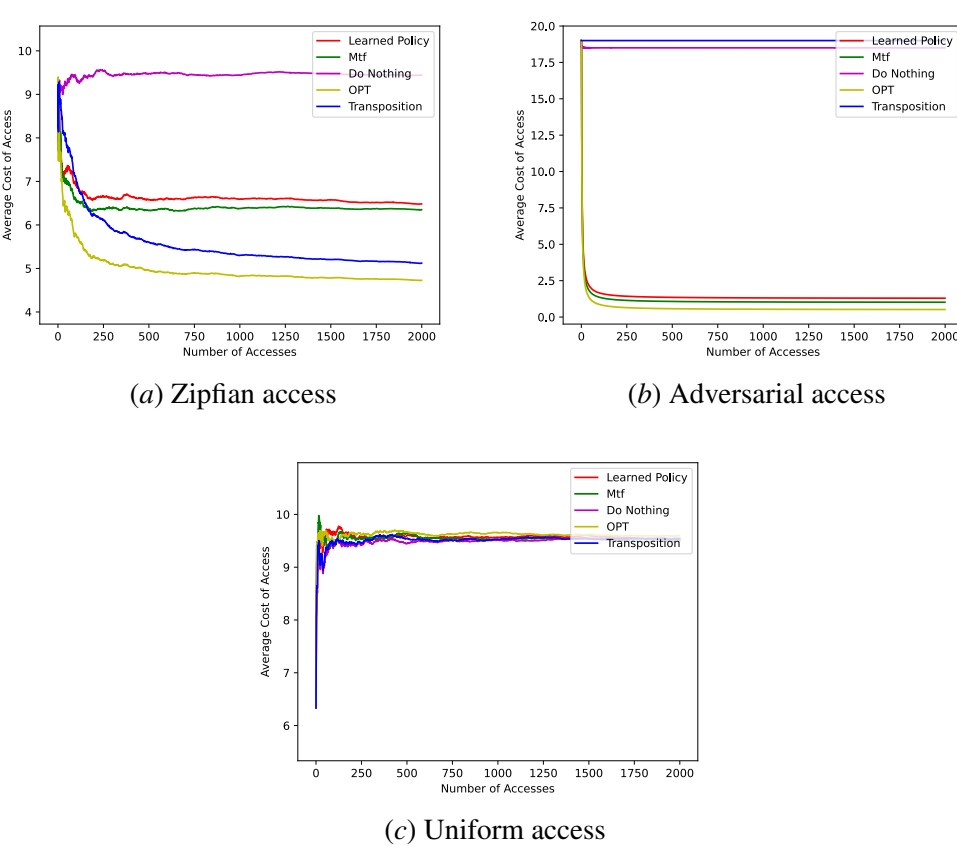

(*a*) Zipfian access           (*b*) Adversarial access

(*c*) Uniform access

Figure 14: Average cost for a list of size 20 on different access sequences.

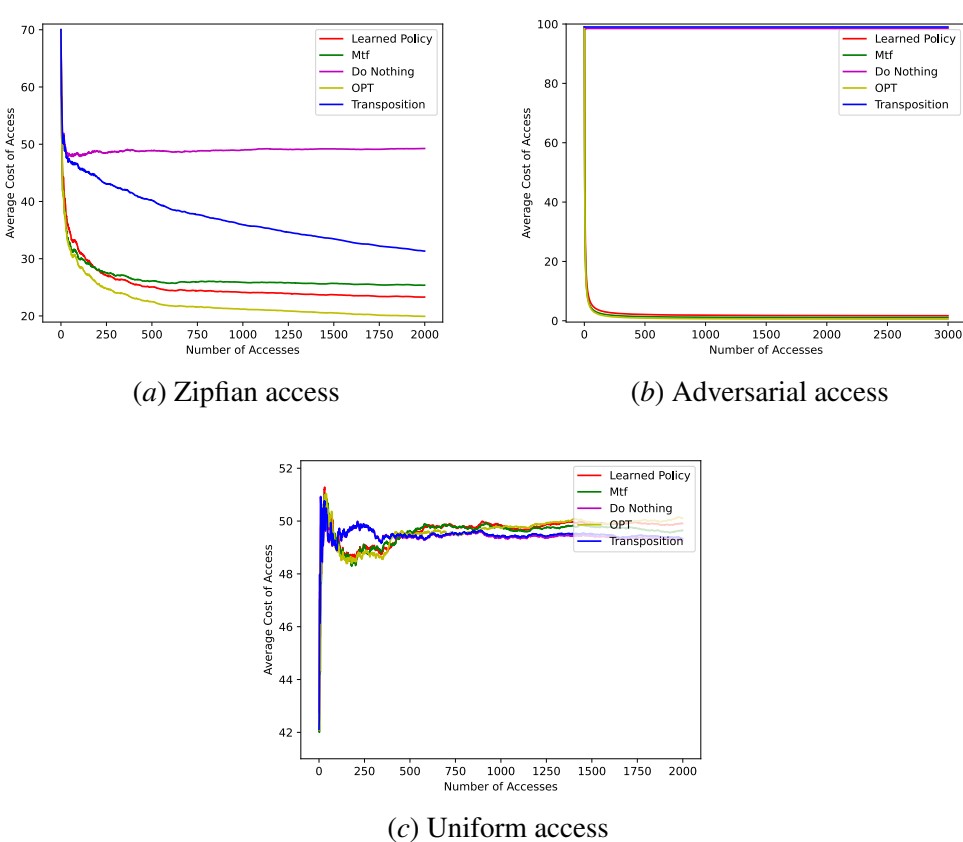

(*a*) Zipfian access                                       (*b*) Adversarial access

(*c*) Uniform access

Figure 15: Average cost for a list of size 100 on different access sequences.

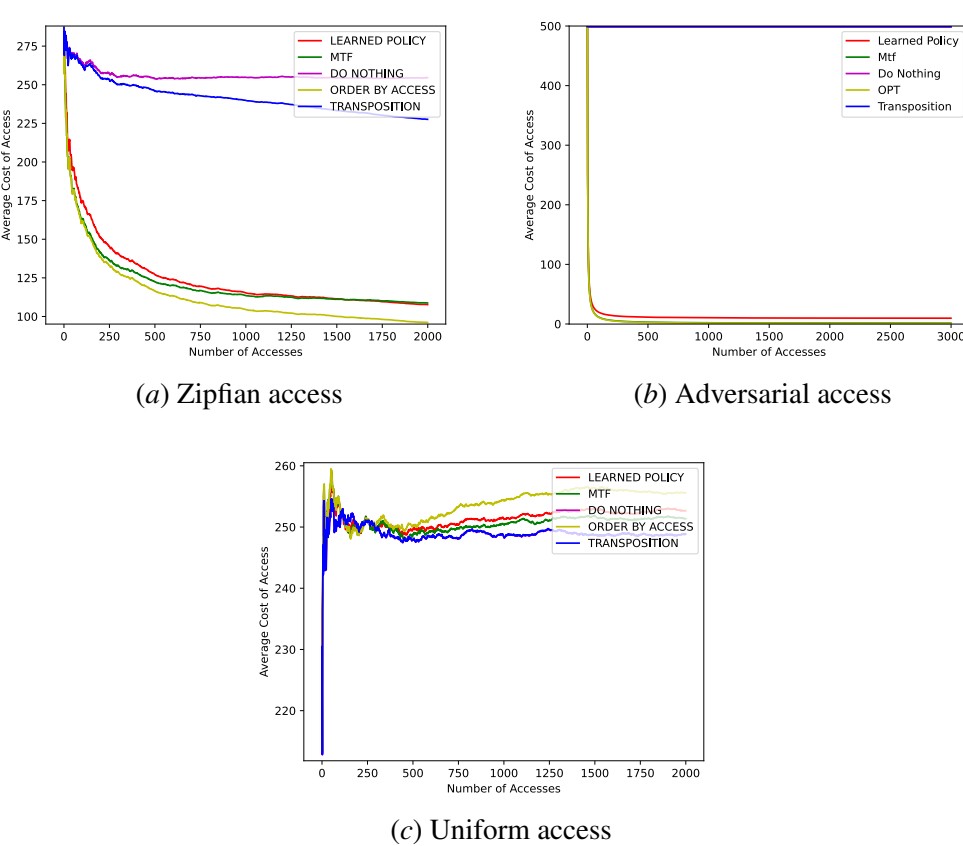

(*a*) Zipfian access

(*b*) Adversarial access

(*c*) Uniform access

Figure 16: Average cost for a list of size $500$ on different access sequences.

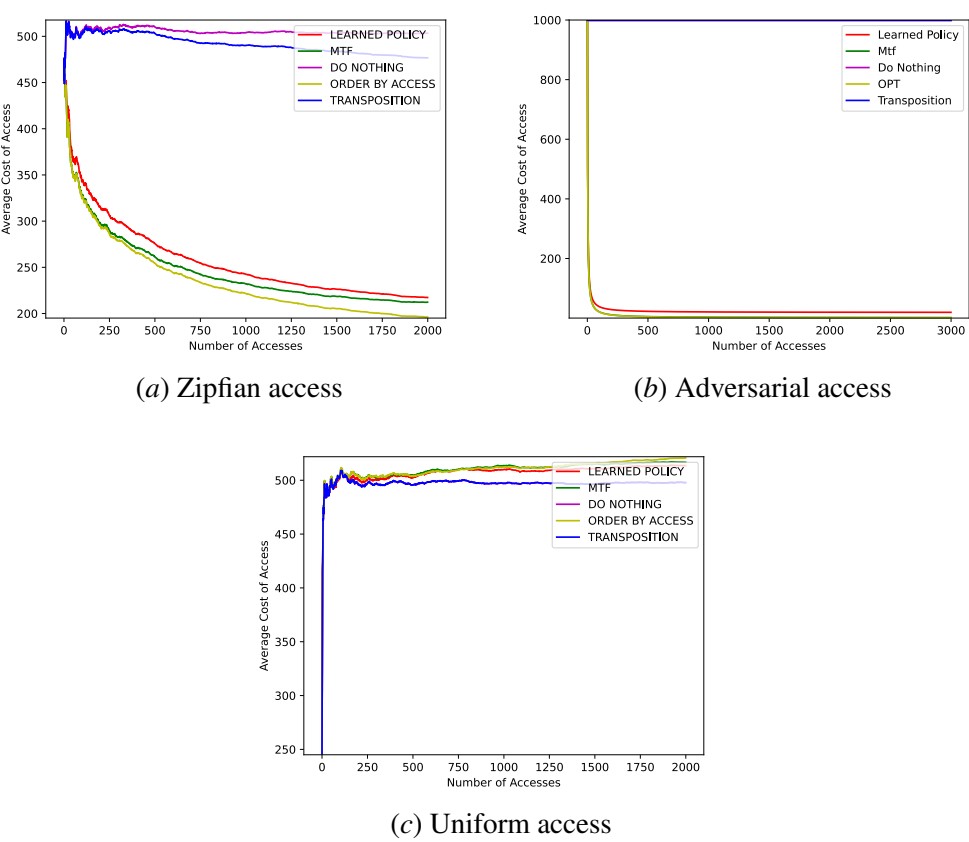

(*a*) Zipfian access  (*b*) Adversarial access

(*c*) Uniform access

Figure 17: Average cost for a list of size 1000 on different access sequences.

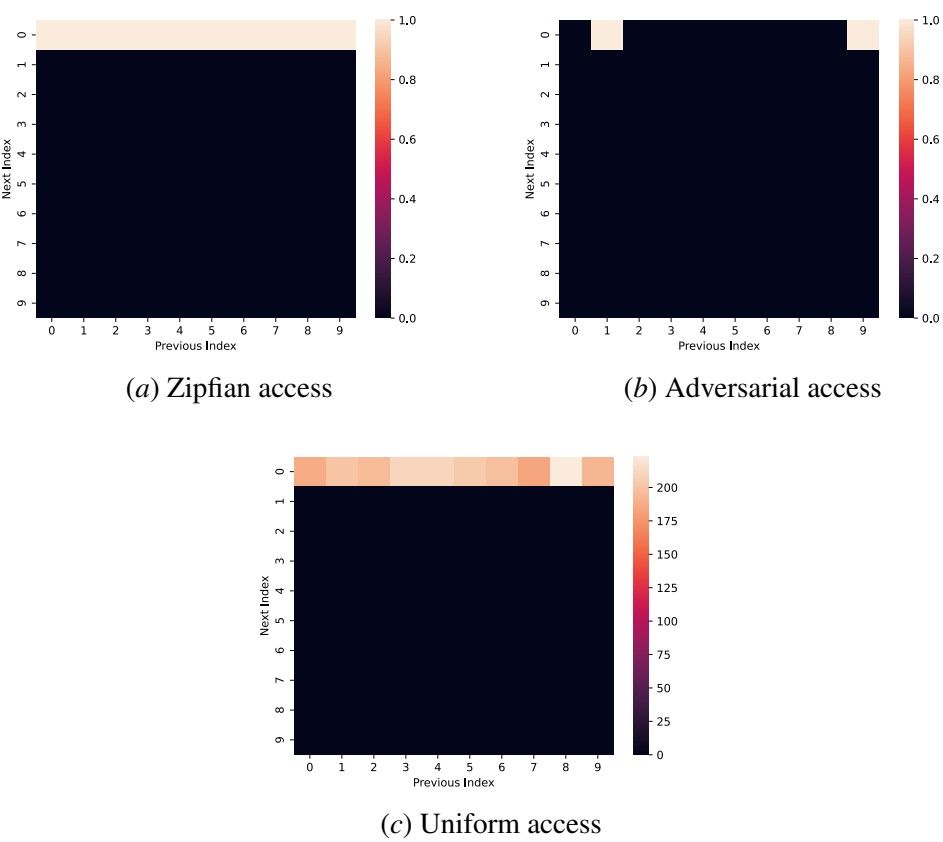

(*a*) Zipfian access

(*b*) Adversarial access

(*c*) Uniform access

Figure 18: Move-To-Front Policy Map for list of size 10 on different access sequences.

