# OpenReview forum: "Optimal and learned algorithms for the online list update problem with Zipfian accesses"
_algorithmiclearningtheory.org/ALT/2025/Conference — ALT 2025_

### Official Review · Reviewer_DSqn · 2024-10-29
**Interesting paper, I recommand accept.**

**Rating:** 7
**Confidence:** 4

**Review:**

This paper studies the problem of online list updates. Here, we have a list of $n$ elements and need to maintain a permutation of the elements at each time step. A query arrives at each time step in the form of an element from the list, and the cost is determined by the rank of this element in the permutation maintained by the learner. The learner then updates the permutation as more queries are observed, with the goal of minimizing the total costs.

This paper specifically studies the case when the queries are sampled i.i.d. from an unknown Zipfian distribution, $\text{Zipf}(\alpha)$, over $[n]$, where $p^{\alpha}[i] \propto \frac{1}{i^{\alpha}}$. The paper shows that the (expected) competitive ratio approaches $1$ as $n \to \infty$ compared to an offline algorithm that knows $p^{\alpha}$. Moreover, this result is achieved by the Transposition algorithm introduced by Rivest (1976). Finally, the paper also introduces an empirical approach based on deep reinforcement learning and provides evaluations of the proposed method.

I have reviewed most of the proofs (though I did not check the detailed algebra carefully in the appendix), and they appear sound. I find the paper interesting and recommend acceptance.


**Strengths**
Although this paper is not directly within my field of expertise, I find the proof ideas quite interesting. The proof relies on a key technique (Lemma 1) from Rivest (1976) for characterizing the stationary distribution of the Markov process induced by the algorithm, together with some novel ideas for a more fine-grained characterization of the stationary distribution.

The paper also mentions that the derived result resolves a conjecture of Rivest (1976), though I cannot comment on its significance as I am unfamiliar with the relevant literature.

**Weaknesses**
1. It seems that the guarantee requires the algorithm to run for a long time to ensure that the Markov chain mixes. Are there any bounds on the mixing time of the underlying Markov chain?
2. It is mentioned that Kan and Ross (1980), Tenenbaum and Nemes (1982), and Gamarnik and Momcilovic (2005) also studied Transposition under i.i.d. accesses. Can the authors comment on how their results differ from yours?
3. There are numerous typos, too many to list here. I recommend the authors conduct thorough proofreading to correct them.

**Paper Award:**

No

---

> ### Author Response · Authors · 2024-11-21
> **Response to Reviewer DSqn**
>
> Thank you for your feedback.
>
> > Weakness 1:  It seems that the guarantee requires the algorithm to run for a long time to ensure that the Markov chain mixes. Are there any bounds on the mixing time of the underlying Markov chain?
>
>
> Thank you for your feedback. There have been some works on bounding the mixing time for the underlying Markov chain if the policy is move to front [1]. However to the best of our knowledge, no such works exist for the harder transposition policy case. We say ‘harder’ because the techniques used in [1] are not directly applicable. They crucially rely on the fact that we can couple to the stationary distribution as soon as all the elements are requested at least once, since after an element is moved to the front its position can always be tracked explicitly.
>
> Thus, different techniques are likely to be required to bound the mixing time of transposition, even for special accesses such as Zipfian. This is an excellent direction for future work.
>
> [1] Fill, James Allen. "Limits and rates of convergence for the distribution of search cost under the move-to-front rule." Theoretical Computer Science 164.1-2 (1996): 185-206.
>
>
> > Weakness 2: It is mentioned that Kan and Ross (1980), Tenenbaum and Nemes (1982), and Gamarnik and Momcilovic (2005) also studied Transposition under i.i.d. accesses. Can the authors comment on how their results differ from yours?
>
> Kan and Ross study this behavior of transposition under a simplistic distribution where $p_2 = p_3 = \cdots = p_n$. Since n-1 elements are identical, their analysis boils down to only tracking the first element. Tenebaum and Nemes extend the analysis for the same access distribution to algorithms that can make $k$ transpositions rather than one.
>
> Gamarnik and Momcilovic study the transposition rule for power law distributions. They prove that transposition ‘sorts’ most of the list in a certain limiting sense: In their setting $p_i \propto 1/i^{\alpha}$ but $\alpha$ must be  $> 1$. $\gamma$ is a fixed constant $< 1$. Let $C$ be the random variable that denotes the search cost in the stationary distribution (the randomness is over the items chosen to be accessed) and let $P$ denote the distribution of the $p_i$’s. They show that if $x < \gamma n$, then  $\lim_{x,n \rightarrow \infty} \frac{\log(\Pr(C > x)) }{\log( \Pr(P > x) )} = 1$. This theorem can be thought of as saying ‘the probability that the search will take a long time is roughly equal to the probability of drawing an item with small $p_i$”, i.e., the logarithm of the tail probability of the search cost is asymptotically optimal.
>
> This by itself is not enough to bound the competitive ratio of transposition for power law distributions up to $1+o(1)$. For example in the Zipfian case of $p_i \propto 1/i$, each element contributes equally to the expected cost (since $i \cdot 1/i$ is constant). The above theorem ignores a constant fraction of the elements (those that are larger than $\gamma n$), so it already cannot track a constant fraction of the expected cost. Taking logs of the two probabilities can also suppress constant factor differences which can impact the competitive ratio. In addition, their theorem only holds for $\alpha > 1$, whereas our main technical work is developed for the $\alpha = 1$ case.
>
> In summary, to the best of our knowledge, the results of prior work do not bound the competitive ratio of Transposition for Zipfian accesses. We are happy to include this discussion in the updated version of the paper.
>
> > There are numerous typos, too many to list here. I recommend the authors conduct thorough proofreading to correct them.
>
> Thank you, we will address the typos in the updated version of the paper.

---

### Official Review · Reviewer_19Go · 2024-11-08
**Review Submission 147**

**Rating:** 6
**Confidence:** 4

**Review:**

The paper considers the well-studied list update problem, where we are given a list of items, and upon the arrival of a requested item, we rearrange the ordering to minimize access cost. The main result of the paper is to establish that Transpose (a well-known online algorithm that exchanges the requested item with the item immediately to its left) achieves a 1+o(1) competitive ratio when requests follow an i.i.d. Zipfian distribution. Additionally, the authors provide interesting experimental evaluations of Deep RL algorithms for the list update problem.

Strengths

The list update problem is a fundamental problem in online algorithms. The results presented are interesting and appear to answer a conjecture posed in 1976. The authors provide a clear proof sketch, with the proofs being non-trivial and technically sound.  At the same time, the experimental results that the authors provide are interesting and I also liked the idea of trying to interpret the behavior learned by the Q-Learning algorithms.

Weaknesses

I believe that the significance of the results is somewhat limited, as they pertain specifically to the stochastic setting for niche family of distributions.

Overall Evaluation

I overall enjoyed reading the paper. I believe it provides interesting results and techniques. Despite the weaknesses that I mentioned above, I believe it should be considered for acceptance.

**Paper Award:**

No

---

> ### Author Response · Authors · 2024-11-21
> **Response to Reviewer 19Go**
>
> Thank you for your feedback.
>
> > I believe that the significance of the results is somewhat limited, as they pertain specifically to the stochastic setting for niche family of distributions.
>
>  The Zipfian distribution is a popular distribution for modeling real world frequency data such as search query distribution or word frequencies. Indeed, many of the empirical papers we could find related to the list update problem explicitly consider the Zipfian case [1-4].
>
> Besides practical applications, the case of Zipfian accesses in the list updated problem was explicitly studied by Ron Rivest [4] and Donald Knuth [5]. Based on empirical evidence in [5], Rivest explicitly asked if Transposition achieves a competitive ratio close to 1 for the Zipfian case. Our paper resolves this 48 year old conjecture and shows that indeed, the competitive ratio of transposition is $1+o(1)$ where $o(1)$ goes to 0 as the list size grows.
>
> Lastly, we believe our work presents novel understanding of the transposition algorithm and is the first to explicitly bound transposition’s competitive ratio for a non-trivial access distribution. We envision that our techniques such as bounding the probability mass of 'very unsorted' lists by those of related 'sorted' lists (Lemmas 2-4) could be applicable in understanding transposition for other general classes of distributions.
>
> Given the reasons outlined above, we believe the Zipfian case to be quite motivated to study.
>
>
> [1] Hester, James H., and Daniel S. Hirschberg. "Self-organizing linear search." ACM Computing Surveys (CSUR) 17.3 (1985): 295-311.
>
> [2] Bachrach, and Reinstädtler. "On the competitive theory and practice of online list accessing algorithms." Algorithmica 32 (2002): 201-245.
>
> [3] Mohanty, Rakesh, and N. S. Narayanaswamy. "Online algorithms for self-organizing sequential search-a survey."
>
> [4] Bachrach, Ran, and Ran El-Yaniv. "Online list accessing algorithms and their applications: Recent empirical evidence." Proceedings of the eighth annual ACM-SIAM symposium on Discrete algorithms. 1997
>
> [5] Rivest, Ronald. "On self-organizing sequential search heuristics.
>
> [6] Knuth, Donald E. The Art of Computer Programming: Fundamental Algorithms, Volume 1

---

### Official Review · Reviewer_Z5ct · 2024-11-09
**Review of ALT 2025 submission 147**

**Rating:** 5
**Confidence:** 3

**Review:**

The list update problem is a very old problem in data structures, asking for an algorithm to maintain a set of items stored as a linked list (in arbitrary order) while answering queries that seek to access items in the list. The list may be reordered in response to the queries, and efficient algorithms for list update reorder the list so that the frequently accessed items appear near the beginning. The Move-To-Front algorithm (MTF) was shown to be 2-competitive by Sleator and Tarjan in a seminal discovery that gave birth to the study of competitive online algorithms. The list update problem under an i.i.d. access pattern was studied by McCabe, and Rivest (1976) showed that the Transposition algorithm (which swaps the accessed item with its predecessor) is at least as good as MTF under every i.i.d. access pattern, in terms of average cost.

The present submission proves a conjecture of Rivest, that the Transposition algorithm is asymptotically optimal (its average cost exceeds the optimum by only a 1+o(1) factor as list size tends to infinity) for Zipfian access distributions. The proof consists of assuming without loss of generality that items are numbered in decreasing order of access frequency, and showing that when j is significantly greater than i — say, when $i$ is at least $\log n$ and $j$ is greater than $i + i^{0.9}$ — then the probability that $j$ appears significantly earlier than $i$ (i.e., $i^{0.15}$ or more positions before $i$ in the list) is negligible, when the ordering is sampled from the stationary distribution of the Transposition rule. This is accomplished by a carefully-constructed accounting system in which states violating the condition are mapped to states satisfying the condition, in a way that allows for comparison of probabilities of the two state sets.

The second main contribution of the paper is a set of experimental results on using deep Q-learning for optimizing over list update policies. I found the experimental setup described in Section 4 to be ill-motivated and potentially riddled with errors. I would appreciate if the authors could address, in their response, the following issues that seemed problematic to me.
1. The paper describes a quadratic state space in which the state at time $t$ is the pair $(r_t,L(r_t))$ consisting of the requested item and its position in the list. The paper claims that under this state representation, the state transitions are Markovian, i.e. the probability of transitioning to $(r_{t+1},L(r_{t+1}))$ given the full past history is equal to the probability of transitioning to $(r_{t+1},L(r_{t+1}))$ given that the previous state was $(r_t, L(r_t))$. I believe that's true for trivial policies that never reorder the list, but false for policies that reorder the list. For example, let's consider the MTF policy and a 3-element list with items {1,2,3}, each with equal access probability. Then $\Pr((r_2,L(r_2)) = (1,2)  | (r_1,L(r_1)) = (2,2),  (r_0, L(r_0)) = (3,3)) = 0$; if 3 is requested at time 0 and 2 is requested at time 1, then the list ordering under the MTF policy will be 2,3,1 at time 2, so if item 1 is requested at that time it will not be found in position 2. On the other hand, $\Pr((r_2,L(r_2)) = (1,2)  | (r_1,L(r_1)) = (2,2), (r_0, L(r_0)) = (1,3)) = 1$; if the first two requests are 1 at time 0 and 2 at time 1, then the list ordering under the MTF policy will be 2,1,3 at time 2, so if item 1 is requested at that time it will certainly be found in position 2. This indicates that the state transitions are not Markovian as claimed.
2. You say that $\max(j,i)$ is the agent's reward in step $t$ if $x$ is queried, found in position $i$, and moved to position $j$. I assume the training objective is to _minimize_ the expected value of $\max(i,j)$, not to maximize it. If so, I think "reward" is a confusing term and should be changed to "cost".
3. I'm confused why you are using deep Q-learning rather than tabular Q-learning given that the state space is only quadratic in $|L|$, and $|L|$ in your experiments does not exceed 1000. For Markov decision processes at this scale, the computational resources required for tabular Q-learning would not be prohibitive.
4. Test sequences of length 3000 seem quite short, when L=500 or L=1000. The mixing time of the Markov chain is at least linear in L, so I would expect the policy's average cost on a test sequence of length 3000 to exhibit strong correlation with the initial condition. Do you have evidence that the average costs observed on a length-3000 test sequence are representative of the long-term average, even when L=500 or L=1000?

My evaluation is that the paper is not appropriate for ALT. There are two aspects to this evaluation.
* The theoretical contribution sketched in Section 3 and detailed in Appendix A. I appreciate the resolution of Rivest's conjecture about the performance of the Transposition policy under Zipfian access distributions. These results resolve an old open problem about average-case analysis of list update policies and deserve publication somewhere. I question whether ALT is an appropriate venue because I do not really see any connection to learning theory. One could say that the Transposition algorithm is "learning" the access distribution by reordering the list, but to me it seems this stretches the definition of learning theory to the point where the average-case analysis of any algorithm whatsoever would be considered "learning theory" if the analysis showed that the algorithm achieved near-optimal performance on its input distribution. To wit, the Transposition algorithm is completely memoryless and encodes no information about learned parameters of the access distribution. (To the extent that the algorithm is "learning" such information, its knowledge is encoded in the state of the list itself, because the Transposition algorithm has no internal state.)
* I am doubtful of the experimental contributions in Sections 4 and 5 for the reasons detailed earlier in this review.

UPDATE AFTER AUTHOR RESPONSE: The authors acknowledged an error in Section 4 and provided me with some important context about Section 5, raising my evaluation of both sections. I am raising my score accordingly.

**Paper Award:**

No

---

> ### Author Response · Authors · 2024-11-21
> **Response to Reviewer Z5ct: Experiments**
>
> > The paper describes a quadratic state space in which the state at time t is the pair (r_t, L(r_t)) consisting of the requested item and its position in the list.
>
> Thank you for your review. Let us explain the choice of "state" representation. Ideally, the states of this process would be the permutations of the list. Using the same example you give of a list of 3 items ABC, that would mean 6 different states- ABC, ACB, BAC, BCA, CAB and CBA. The transitions from one state to the other for any list update policy we can think of  will be Markovian. Unfortunately, using this state representation and computing the Q-function is intractable as the size of the list grows, since that requires $|L|!$ many states.
>
>  Therefore, we devise the following work around: we apply a transformation to the states to reduce the state space from $O(|L|!)$ to $\text{poly}(|L|)$ so we can compute the Q-function empirically. To do this, we considered multiple transformations and chose the transformation described in the paper as it actually allows the agent to learn a policy that was competitive with OPT.
>
> This transformation takes the state space from $|L|!$ to $|L|^2$ since there are $|L|$ possible items that could be queried and there are $|L|$ possible places this item could be. Thus, anytime we want to look up the optimal action based on our current list state i.e. (ABC, ACB, BAC, BCA, CAB and CBA for a list of size 3 for example), we first pass this list state and the current observation to our transformation function. This would then give us a new state in the transformed state space which we then look up in the Q-table. Our argument then, is that this transformation should not affect the Markovian nature of the process itself. In effect, we are approximating the true Q-function which would be a $|L|! \times |L|!$ table with a $|L|^2 \times |L|^2$ one. We hope that this clarifies any misunderstanding.
>
> > You say that max(j, i) is the agent's reward in step t ... If so, I think "reward" is a confusing term and should be changed to "cost".
>
> The reward is $-\max(j,i)$ which we maximize. This is standard in reinforcement learning (e.g. see Chapter 3 in [1]).
>
> [1]: Reinforcement Learning an Introduction, 2nd Edition, Richard S. Sutton and Andrew G. Barto
>
> > I'm confused why you are using deep Q-learning rather than tabular Q-learning
>
> Thanks for this question. Indeed, given that state description, there is no need for us to consider deep Q-learning. However, let us refer back to the description of our procedure above. Since we aim to approximate an $L! \times L!$ function with a $L^2 \times L^2$ function, tabular Q-learning may not be appropriate because this is a much more complicated approximation. Now suppose we apply the compression function and then perform tabular Q-learning, in the following figure https://ibb.co/8xXBcnH (uploaded to an anonymous repository) you can see the learned policy and the performance of the agent is extremely poor, even for a list size of 50. This is because tabular Q-learning is not powerful enough to approximate the Q-function for the states given the transformation we apply. Therefore, we need a more complex approximator hence why we use a neural net for learning the Q-function. We hope this clarifies any misunderstanding.
>
> > Test sequences of length 3000 seem quite short, when L=500 or L=1000. The mixing time of the Markov chain is at least linear in L, so I would expect a test sequence of length 3000 to exhibit strong correlation with the initial condition. Do you have evidence that the average costs observed on a length-3000 test sequence are representative of the long-term average, even when L=500 or L=1000?
>
> Thank you for your observation. The convergence time depends on the algorithm since each defines a different Markov chain. As we mention in the Appendix of the paper and as can also be seen in Zipfian plots for list size > 20, transposition, while competitive takes a long time to reach stationary and so even after 3000 queries in a list of size 50, it still has not stabilized. The convergence time may be faster for other algorithms and our learned policy. However, the convergence time is not the main focus of our experiments. Rather, we want to show that our learned RL based algorithm is competitive with the best classical algorithms across a variety of access patterns and time scales. This is clearly demonstrated by our experiments: our learned policy is good across Zipfian and adversarial accesses, whereas transposition can be easily “hacked” by carefully breaking the i.i.d. assumption. The quality of our learned algorithm remains high even as we increase the number of accesses. Indeed in the following figure https://ibb.co/v1fDsky (uploaded to an anonymous repository), we increased the number of access to 10,000 for a list of size 500. The results are qualitatively similar to the figures in our paper.

---

> > ### Author Response · Authors · 2024-11-21
> > **Response to Reviewer Z5ct: Fit for the conference**
> >
> > > The theoretical contribution sketched in Section 3 and detailed in Appendix A. I appreciate the resolution of Rivest's conjecture about the performance of the Transposition policy under Zipfian access distributions. These results resolve an old open problem about average-case analysis of list update policies and deserve publication somewhere. I question whether ALT is an appropriate venue because I do not really see any connection to learning theory. One could say that the Transposition algorithm is "learning" the access distribution by reordering the list, but to me it seems this stretches the definition of learning theory to the point where the average-case analysis of any algorithm whatsoever would be considered "learning theory" if the analysis showed that the algorithm achieved near-optimal performance on its input distribution. To wit, the Transposition algorithm is completely memoryless and encodes no information about learned parameters of the access distribution. (To the extent that the algorithm is "learning" such information, its knowledge is encoded in the state of the list itself, because the Transposition algorithm has no internal state.)
> >
> > We argue that our results fall under the umbrella of learning theory due to the following points.
> >
> > - (1) In the context of memory-less list update algorithms, any competitive algorithm must inherently "learn" key properties of the underlying access distribution. For instance, it must identify and prioritize frequently accessed elements, the "heavy hitters", to optimize the list's order. This can be viewed as a form of memory-limited learning, where the algorithm is not allowed to explicitly remember any samples! Nevertheless, it intuitively learns to encode relevant distributional information by rearranging the list. We also remark that memory-limited learning in other contexts, e.g. streaming algorithms, is a popular research topic (e.g. see [1] and references therein) and we believe our paper contributes to this line of work.
> >
> > - (2) Our work highlights a different perspective on distribution learning, focusing on optimizing the search cost rather than minimizing traditional metrics like total variation distance. In other words, we ultimately want our algorithm to learn to organize to support fast searches. Of course learning the access distribution up to high accuracy (low total variation (TV) distance) is sufficient, but not necessary. This distinction of learning 'enough' for a downstream task, rather than learning the distribution outright, is present in many other classic learning theory problems, for example many problems in improper learning. This perspective can introduce new research directions, highlighted in point (3) below.
> >
> >
> > - (3) We believe our work also has connections to two classic topics under learning theory: online algorithms and understanding Markov chains. The first topic is clear so we expand upon the second. Our paper studies an interesting Markov chain that is practically motivated: the states are all possible list permutations and the transition probabilities between states are defined by the access distribution and the transposition algorithm. As pointed out by other reviewers, our work leads to many exciting future questions in this direction: How fast does the Markov chain converge? Is there a distinction between the Markov chain converging to the stationary up to small TV versus converging to make the search cost within $1+o(1)$ of OPT? The answers may be different because the search cost is weighted by the position of the items. Is there a separation between learning the access distribution and optimizing the search cost? One formulation of this is: we know there is a necessary sample complexity needed to learn the access distribution up to TV distance $\epsilon$. How large can the search cost be (as a function of  $\epsilon$) if we sort according to a worst-case distribution that is within TV distance epsilon of the true access distribution? Does the Markov chain converge to a “good solution” (e.g. a similar search cost as the previous bound) in much fewer steps than the sample complexity of learning the access distribution up to TV  $\epsilon$? We believe these are all questions interesting to the broader learning theory community.
> >
> > Lastly, we remark that our actual learning experiments inspired our theory: We observed that the RL algorithm is very conservative in its movement (see Figure 3a in our paper). This motivated us to prove a result for the most conservative algorithm, transposition, which interestingly was also raised as a candidate for optimality by Rivest.
> >
> > [1] Hypothesis Selection with Memory Constraints. Aliakbarpour, Bun, Smith. NeurIPS 24

---

> > ### Comment · Reviewer_Z5ct · 2024-12-01
> > **Response does not acknowledge error**
> >
> > I read the authors' response to my review, but their response didn't satisfy my concerns. The main disagreement is about Section 4 on page 9.
> >
> > Sections 4 and 5 are experimentally evaluating policies for the online list update problem, i.e. the problem of maintaining a linked list $L$ storing a set of $n$ elements, reordering the list as necessary so as to minimize the cost of accessing a sequence of requested elements. For a stochastic stream of requests with known distribution, the problem can be described as an MDP with $n!$ states representing the possible orderings of the list. The paper instead uses a "state space" of size $n^2$ where the meaning of state $(i,j)$ is that item $i$ was requested at a time when it was located in the $j^{\mathrm{th}}$ position of the list.
> >
> > In my review I pointed out that the dynamics are not Markovian with respect to this state-space representation, and I even presented an example to show that the probability of transitioning from state $(i,j)$ to $(i',j')$ in the quadratic-sized state representation can be history-dependent. The authors' response, as I understand it, points to the intractability of representing the full $n!$-sized state space as a justification for using $n^2$ states. I acknowledge that representing the full $n!$-sized state space is not computationally feasible; my issue is that the paper erroneously asserts that the state transitions in the quadratic-sized state representation are still Markovian. This assertion is even accompanied by an equation (the displayed equation in the middle of page 9) asserting that the probability of transitioning from state $(r_t,L(r_t))$ to $(r_{t+1},L(r_{t+1}))$ does not depend on the history of states visited prior to time $t$. The counterexample in my review falsifies this equation, and the authors' response neither acknowledges the mistake nor explains the error in my reasoning.

---

> > > ### Author Response · Authors · 2024-12-01
> > > **Response to Reviewer Z5ct**
> > >
> > > Dear Reviewer Z5ct,
> > >
> > > Thank you for the clarification. We now understand what you mean. Indeed, we apologize for this error on page 9. We intended to state that if one used the full $n!$ sized list as part of our state, then the states would be Markovian. As you correctly pointed out, this is not experimentally tractable. We will update this in the next version. However, as stated in our earlier response, one can still hope that the quadratically many states we are using are `expressive enough' in practice. Indeed, we empirically demonstrate that our state choice is powerful enough to learn good online policies across a range of access patterns in our experiments.
> > >
> > > We believe we have also addressed your other concerns:
> > >
> > > - Fit for the conference (Comment https://openreview.net/forum?id=zWpqMU5Vqc&noteId=3LKixNXzPd)
> > > - Advantage of deep Q learning in our experiments as opposed to tabular Q learning (see the additional experiments we did in https://openreview.net/forum?id=zWpqMU5Vqc&noteId=OCaCOEgxR4)
> > >
> > > Lastly, we would like to point out that our main contribution to the paper is a theoretical result resolving an old question of Rivest. Our experiments demonstrate that RL can actually learn good online policies, which is the main point we are trying to argue in our empirical section. Indeed, as we stated in a comment above, the behavior or the learned policy actually inspired us to prove Rivest's conjecture on transposition.

---

> > > > ### Comment · Reviewer_Z5ct · 2024-12-01
> > > >
> > > > Thank you, your follow-up to my comment is helpful and I think I'm approaching a better understanding of the motivation behind the setup in Sections 4 and 5 of your paper. There's one thing that still puzzles me, however, and I'm curious if you can shed some light on it.
> > > >
> > > > In justifying the choice to reduce the state space from $|L|!$ size to $|L|^2$ size, you pointed out that representing the true Q-function as a table of size $|L|!$-by-$|L|!$ would be intractable. I fully agree that representation would be intractable. However, in your experiments in Section 5, rather than representing the Q-function as a table, didn't you represent an approximation of the Q-function as a neural network with $2|L|$ input units, by encoding a pair $(r_t,L(r_t))$ as a pair of 1-hot encodings? If so, I wonder if you thought about the option of representing the entire list $L$ in the input layer of the network? This could be done with $n^2$ input units, by encoding the list $L$ as an $n \times n$ matrix: either the permutation matrix with $a_{ij} = 1$ if and only if $L(i) = j$, or the comparison matrix with $a_{ij} = 1$ if and only if $L(i) \geq j$. This would entail a quadratic increase in the size of the input layer of the neural network, much milder than the factorial increase we've been discussing in this thread. And it would come with the benefit that the network architecture, at least in principle, could approximate the Q-function on the full state space rather than the reduced quadratic-sized state space you used in your experiments. Is there a reason you chose instead to work with a neural network whose input layer corresponded to the quadratic-sized reduced state space?

---

> > > > > ### Author Response · Authors · 2024-12-02
> > > > > **Response to Reviewer Z5ct**
> > > > >
> > > > > Thank you for your comment. Yes, we tried a variety of different state representations, including the list as a permutation matrix. In our experiments, we did not notice any significant advantages of using alternate representations. For example in this figure (https://ibb.co/n1CBzLh, anonymous image repository), we tried many options which are described at the end of the comment.
> > > > >
> > > > > Explanation of the figure:
> > > > >
> > > > > The figure is for experiments on a list of size 50 with the Zipfian distribution. We show the average cost per episode, where an episode represents 5000 accesses fixing the distribution and list. In an new episode, we change the distribution and list completely (we use a Zipfian distribution on a totally new list). This is similar to our experiments in Appendix B (also described in the beginning of Section 5). The choice of the paper is the 'CURR_POSITION' line. We can see that its performance is clustered around a few other choices, but definitely superior to the permutation matrix choice. Note that for all of these, we fixed the neural network, except the input layer.
> > > > >
> > > > > It is not entirely clear to us why the permutation state is not performing as well as the other choices we tried. This maybe due to the fact that our network was not big enough to fully take advantage of the more powerful representation. A bigger network or many more training iterations maybe required to fully learn over the factorial many possible inputs. However, this was not exactly the main focus of our experiments. Rather, we sought to show that for the classic list update problem, a RL algorithm can actually learn sensible and interpretable policies that are competitive for a wide range of accesses. We leave a deeper investigation of different state choices as an interesting direction for future empirical work.
> > > > >
> > > > > Description of the different states that we tried:
> > > > > - Permutation matrix: Encode the list exactly as a permutation matrix.
> > > > > - First Item: The first record in the list $L$.
> > > > > - Preceding Item: Given that access $r_t$ references the record in position $i$, the preceding item list state representation chooses the record $i-1$ as the list state. If $i=1$, then the preceding item list state chooses the record $i$ itself.
> > > > > - Succeeding Item: Given that record $r_t$ queries the record in position $i$, the succeeding item list state representation chooses the record $i+1$ as the list state. If $i=|L|$, then the succeeding item list state chooses the record $i$ itself.
> > > > > - Preceding and Succeeding Item: This list state representation uses both the preceding and succeeding item representation. Note that the size of the total state space is $O(|L|^3)$, including the requested item.
> > > > > - Current Position: The state choice we used in our main experiments
> > > > > - First $\log$: The first $\log|L|$ records of $L$ are the list state representation. The size of the total state space is $|L|\cdot{|L| \choose \log |L|}$.
> > > > > - Last $\log$: The last $\log|L|$ records of $L$ are the list state representation. Same size as above.
> > > > > - First member of $\log$: Given a list $L$, break it up into $\frac{|L|}{\log |L|}$ sections each of size $\log |L|$. The list state representation consists of the first record of each $\log |L|$ chunk.  This list state representation has size $\frac{|L|}{\log |L|}$ and the total size of the state representation is $\frac{|L|}{\log |L|} + 1$. The size of the total state space is $|L|\cdot{|L| \choose \frac{|L|}{\log |L|}}$.

---

> ### Author Response · Authors · 2024-11-21
> **Response to Reviewer Z5ct**
>
> Dear Reviewer Z5ct,
>
> We have split our response to two comments, one addressing experiments and the other addressing the fit of our problem to the conference. We hope we have clarified your main concerns. If your concerns have not been resolved, could you please let us know which concerns were not sufficiently addressed so that we have a chance to respond?
>
> Many thanks,
> The authors

---

### Meta-Review · Area_Chair_Pddm · 2024-12-14

**Recommendation:** Accept
**Confidence:** 4

**Metareview:**

This paper resolves a conjecture by Rivest, and shows that the transposition algorithm is asymptotically optimal for Zipfian access distributions. From that stand point the paper seems like a clear accept. However, I do agree with a couple of points brought up by one of hte reviewers. While I dont agree with the reviewer that the paper's topic is not suitable for ALT, I do agree with the reviewer that proof turned out to not be very illuminating and does not introduce new proof techniques to the field. It does resolve the conjecture but it is also not clear to me how significant the conjecture was. I also felt that the experimental evaluation of the paper seems a bit tangential. Reading the response of the author gave me some reassurances but I still dont quite understand why inputing the entire list L as input to a neural network doest perform as well. I also felt that the  deep Q-learning part was kind of disconnected to the first part of the paper.

But all this aside, I am still leaning towards an accept.

**Paper Award:**

No